mathematical modelling/health and disease and epidemiology

COVID-19, mathematical modelling, social distancing, healthcare burden, cost analysis, testing impact

**Author for correspondence:**
I. R. Moyles
e-mail: imoyles@yorku.ca

# Cost and social distancing dynamics in a mathematical model of COVID-19 with application to Ontario, Canada

## I. R. Moyles[1,2], J. M. Heffernan[1,2] and J. D. Kong[1,2]

[1]Department of Mathematics and Statistics, and [2]Centre for Disease Modelling (CDM), York University, Toronto, Canada

 IRM, 0000-0003-1127-4085; JMH, 0000-0001-9502-1688;
JDK, 0000-0002-7557-5672

A mathematical model of COVID-19 is presented where the decision to increase or decrease social distancing is modelled dynamically as a function of the measured active and total cases as well as the perceived cost of isolating. Along with the cost of isolation, we define an overburden healthcare cost and a total cost. We explore these costs by adjusting parameters that could change with policy decisions. We observe that two disease prevention practices, namely increasing isolation activity and increasing incentive to isolate do not always lead to optimal health outcomes. We demonstrate that this is due to the fatigue and cost of isolation. We further demonstrate that an increase in the number of lock-downs, each of shorter duration can lead to minimal costs. Our results are compared with case data in Ontario, Canada from March to August 2020 and details of expanding the results to other regions are presented.

## 1. Introduction

As of February 2021, there have been over 106 million cases of COVID-19 worldwide, over 808 000 cases in Canada, and over 284 000 cases in the province of Ontario. The early stages of the outbreak focused on mathematical modelling of disease dynamics such as transmission and the basic reproduction number [1,2].

It quickly became clear that asymptomatic spreading was important and that undetected infections were important to consider in models [3]. This caused a global policy shift towards travel restrictions, community closures and social distancing implementations. The impacts of mathematical modelling on policy are documented in [4].

The implementation of non-pharmaceutical intervention (NPI) such as social distancing quickly became an important mathematical modelling task (cf. [5–8]). The majority of these models focus on fixed policy implementations such as reducing contacts on a given date and reinstating them on another. There are two main issues with this, the first is that it requires knowledge of the implementation and relaxation times. While this can be explored in model simulations and optimized for best results, its independence from the model itself can make it hard to adapt to other diseases, strains, or important factors. A second problematic issue is that it assumes an instantaneous policy compliance, i.e. that people will immediately reduce contacts upon implementation and stop upon relaxation. While this can be impacted by an adherence parameter, it does not allow for a dynamic response which is more realistic of human choice. Therefore, a dynamical social distancing model that reacts to the disease dynamics is more realistic.

A dynamic intervention strategy where intervention was turned on and off based on the state of the epidemic was considered in [8] where a decrease in both total infections and social distancing duration was observed compared with a fixed-duration intervention which they also considered. However, modelling the dynamics of intervention entirely on the disease progression assumes that people will immediately distance or relax at some threshold. This suggests that a periodic solution will emerge centred around the critical disease threshold and this appears to happen in [8]. While it is quite realistic that disease dynamics drive people into isolation, it is a separate mechanism, namely the cost of staying home, that people consider when relaxing their isolation habits. Cost is seldom considered in models, with most of the focus on larger economic influence [9,10]. These economic factors certainly play a role in individual cost but psychological factors such as loneliness and habit displacement are important as well.

For this paper, we propose a differential equation model for the spread of COVID-19 with separate dynamics for isolation and relaxation dependent on disease progression and relaxation cost, respectively. The disease progression information typically comes from media reports and has been investigated in the context of infectious diseases such as influenza (cf. [11–14]) and is usually used to reduce the susceptibility of individuals who are positively influenced by media. The relaxation cost is less often considered and its inclusion recognizes that repeated lock-downs would have diminishing returns as the cost to stay home becomes too overwhelming. A dynamic response model allows for more realistic policy strategies for disease mitigation and mortality prevention. Our model focuses on the spread of the disease in Ontario, Canada, but could be adapted with other parameters to other regions.

Our study is outlined as follows. In §2, we introduce the model and the dynamic response functions for social distancing and relaxation. We also introduce the parameters including those which we fit to data from [15]. We define health, economic and total costs of the pandemic. The health cost is based on overloading existing healthcare resources while the economic cost is the personal or societal cost of social distancing. We show the excellence of fit to our data in §3 and present a series of results based on different scenarios where policy parameters that control distancing and relaxing are varied. We consider scenarios where both health and relaxation costs are equally weighted or where health cost is much more strongly influencing the total cost. We consider a modification to the relaxation rate so that it depends on both cost and cases and see that multiple outbreak peaks can occur. We discuss the implications and conclusions of our work in §4.

## 2. Model

We consider a mathematical model for COVID-19 consisting of classes of people with various exposure to the disease. These classes are listed in table 1 where we note that the removed groups include people who have died from the virus which we do not separately consider. For each of the population classes, we assume there are three levels of social distancing indicated by a variable subscript zero, one or two. If the subscript is 0 then there is no social distancing, subscript 1 indicates that there is social distancing which reduces the contact probability by some percentage while for subscript 2, the contact probability is zero, i.e. full isolation. We introduce a further subscript, $M$ which represents the mitigation of spread due to individuals who have tested positive and are isolated. We assume that only $P$, $I_S$ and $I_A$ populations can test positive and that these people will immediately and completely isolate effectively placing them in the social distance two category for the duration of their disease.

We follow the usual SEIR model framework (cf. [21,22]) which we illustrate in figure 1 with equations detailed in appendix A.

**Table 1.** Variable and parameter definitions

| | definition | value | comment |
|---|---|---|---|
| $S$ | susceptibles, people who can catch the virus | $0.9998N$ | initial condition |
| $E$ | exposed, people who have caught the disease but are not yet infectious | 0 | initial condition |
| $P$ | pre-symptomatic, people who are infectious but have not had the disease long enough to show symptoms | 0 | initial condition |
| $I_S$ | infected-symptomatic, people who are infectious and have started showing symptoms | $2.00 \times 10^{-4}N$ | initial condition |
| $I_A$ | infected-asymptomatic, people who are infectious but never show symptoms | 0 | initial condition |
| $R_S$ | removed-symptomatic, people who were symptomatic and infectious, but are no longer infectious | 0 | initial condition |
| $R_A$ | removed-asymptomatic, people who were asymptomatic and infectious, but are no longer infectious | 0 | initial condition |
| $N$ | population of Ontario | 13 448 494 | 2016 census |
| $N_{crit}$ | critical population at which healthcare resources are overwhelmed | 81 301 | chosen |
| $R_0$ | basic reproduction number | 2.40 | [16,17] |
| $\beta$ | transmission rate of disease after coming in contact with the infected class | $0.223 \text{ d}^{-1}$ | see (2.10) |
| $\delta$ | reduction in transmission due to social distancing in class 1 | 0.250 | chosen |
| $\alpha$ | reduction in transmission due to being asymptomatic | 0.500 | chosen |
| $\sigma$ | rate at which exposed class enter pre-symptomatic class | $2.00 \text{ d}^{-1}$ | [16] |
| $\phi$ | rate at which pre-symptomatic class can begin showing symptoms | $4.60^{-1} \text{ d}^{-1}$ | [16,18,19] |
| $Q$ | proportion of infected individuals who show symptoms | 0.690 | median value |
| $\gamma$ | rate at which an infected person is no longer infectious | $10.0^{-1} \text{ d}^{-1}$ | [20] |
| $\mu_{max}$ | maximal rate at which someone moves from a less socially distant class to a more socially distant class | $1.00 \text{ d}^{-1}$ | chosen |
| $\nu_{max}$ | maximal rate at which someone moves from a more socially distant class to a less socially distant class | $1.00 \text{ d}^{-1}$ | chosen |
| $\mu_I$ | rate at which people showing symptoms choose to isolate | $0.010 \text{ d}^{-1}$ | chosen |
| $q_0$ | proportion of $S_0$ socially distancing into $S_1$ | 0.9 | chosen |
| $q_2$ | proportion of $S_2$ relaxing social distancing into $S_1$ | 0.6 | chosen |
| $q_I$ | proportion of symptomatic individuals $I_{S0}$ who isolate into $I_{S1}$ | 0.6 | chosen |
| $\rho_A$ | testing rate for someone not showing symptoms to test positive | $8.70 \times 10^{-3} \text{ d}^{-1}$ | see appendix B |
| $\rho_S$ | testing rate for someone showing symptoms to test positive | $3.48 \times 10^{-2} \text{ d}^{-1}$ | see appendix B |
| $M_c$ | critical active cases to induce social distancing | $2.09 \times 10^3 / N_{crit}$ | see appendix B |
| $M_0$ | active cases that lead to half the maximal rate of social distancing | $4.18 \times 10^3 / N_{crit}$ | see appendix B |

*(Continued.)*

| | definition | value | comment |
|---|---|---|---|
| $k_c$ | critical approximate disease doubling rate to induce social distancing | $16.2^{-1}$ d$^{-1}$ | see appendix B |
| $k_0$ | approximate disease doubling rate that leads to half the maximal rate of social distancing | $4.06^{-1}$ d$^{-1}$ | see appendix B |
| $C_c$ | critical cost to induce social relaxation | 50 d | chosen |
| $C_0$ | cost that leads to half the maximal rate of social relaxation | 100 d | chosen |

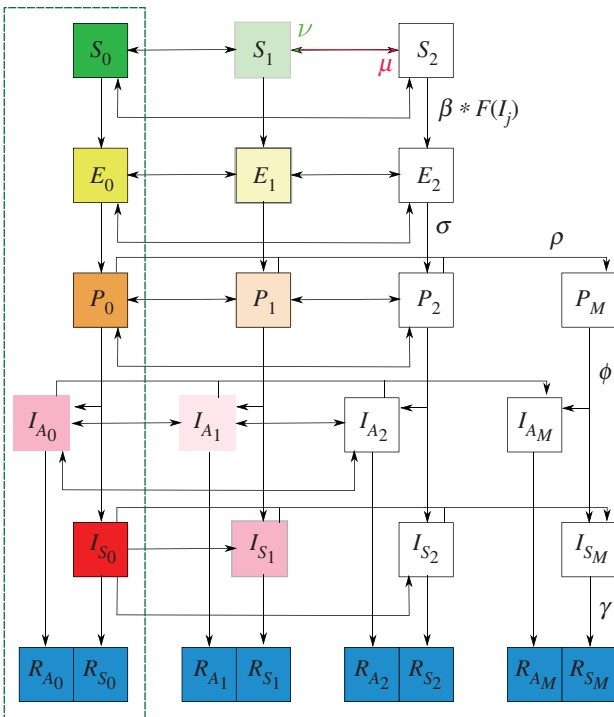

**Figure 1.** Graphical representation of the SEIR model used throughout the manuscript. We use fading to indicate a reduction in transmission which comes from distancing/isolation and also from the asymptomatic disease carriers being less infectious. These effects are quantified by $\delta$ and $\alpha$ respectively in the full model detailed in appendix A. For a condensed graphical representation, we have indicated a representative parameter on particular arrows; however, in general each parameter has a subscript $(i, j)$ indicating the originating and terminal compartment respectively, details of which are in the main text. The parameter $\mu$ (red arrowhead in figure) indicates distancing to a higher category while $\nu$ (green arrowhead in figure) indicates relaxing down to a lower category. $F(I_j)$ represents the force of infection indicating that susceptible people require interaction with one of the infected classes for successful disease transmission. The portion of the model in the green rectangle is the model when social distancing and testing is not considered.

For the model, we assume that vital statistics are not important on the time scales we consider so we take a fixed population $N$. We also normalize the model by another population $N_{\mathrm{crit}}$ which is the amount of people needing healthcare resources that puts the system at full capacity. Figure 1 shows the various model parameters which are summarized in table 1. Each transition parameter has a subscript $(i, j)$ with $i$ the originating class and $j$ the terminal class. The exception to this is $\beta_{i,j}$, where $i$ is the class of the susceptible person and $j$ the class of the infected contact. We make the following assumptions about the model parameters:

(i) the parameters $\delta$, $\sigma$, $\phi$, $\gamma$ and $Q$ are constant and the same for each social distancing class as the disease progression characteristics are unaffected by social distancing. The social distancing partition parameters $q_0$, $q_2$ and $q_I$ are also constant.

(ii) $\beta_{S_0 I_{S_0}} = \beta$ a constant which incorporates both contact and disease transmission probability. We assume that people not showing symptoms shed a lower viral load and hence reduce

transmission by a constant factor $\alpha$ and that those in social distancing class 1 reduce their contacts by a constant factor of $\delta$, effectively also reducing their transmission. For example, $\beta_{S_0 I_{A_0}} = \alpha\beta$, $\beta_{S_1 I_{S_0}} = \delta\beta$ and $\beta_{S_1 I_{A_1}} = \alpha\delta^2\beta$.

(iii) people in the infected symptomatic class $I_{S_0}$ choose to isolate at a constant rate $\mu_I$ with $q_I$ going into $I_{S_1}$ and $(1 - q_I)$ going into $I_{S_2}$. They stay in the social distancing class until they have recovered from the disease, i.e. $\nu_I = 0$. Furthermore, this means that someone already social distancing in state 1 or 2 who becomes symptomatic remains in that social distance class. We note that individuals in $I_{S_0}$, $I_{S_1}$ and $I_{S_2}$ know that they are sick but have not tested positive for the disease. If they test positive, they transition to $I_{S_M}$ and are completely quarantined.

(iv) there are two testing rates $\rho_A$ and $\rho_S$ for asymptomatic (including pre-symptomatic) and symptomatic individuals, respectively, with $\rho_S > \rho_A$ as we assume that symptomatic people are more likely to seek out a test as they have symptoms. Asymptomatic people are likely to only seek a test out if they believe, through contact tracing or otherwise, they have come into contact with someone who has the virus, or through targeted testing initiatives. Despite the fact that testing numbers fluctuate with the progression of the disease, we take the testing rates to be constant which makes them an effective testing rate. This is consistent with studies estimating global infections per symptomatic test case (cf. [23,24]). To help restrict the model, we take $\rho_S = 4\rho_A$ which is a similar value as observed in [24], which compared data from Germany, South Korea and the USA.

(v) we assume that only people in the $P$, $I_S$ or $I_A$ classes will test positive if a test is administered. Therefore, we explicitly assume people in the $E$ class do not have a high enough viral load to shed.

(vi) people who have tested positive are isolated (effectively put in social class 2) until recovery and cannot transmit the disease. This ignores infections to family members living in a household with an isolated member or infections to healthcare workers who are conducting tests or treating COVID-19 patients. See [25] for considerations of a model with household structure included.

## 2.1. Social distancing and testing

Since people without symptoms are unaware of whether they have the virus or not, we assume that both social distancing and relaxing rates are independent of the disease class they are in. We therefore define $\mu$ as the rate of social distancing from state 0 with proportion $q_0$ going to state 1 and $(1 - q_0)$ going to state 2. We similarly define $\nu$ as the rate of decreasing social distance from state 2 with proportion $q_2$ going to state 1 and $(1 - q_2)$ going to state 0. We define those social distancing from state 1 to state 2 as $\mu/2$ to account for the fact that anybody in state 1 has already undergone one transition and so they should be slower at making a secondary transition. For a similar reason, we define the social distancing relaxation from state 1 to state 0 as $\nu/2$.

Testing provides two important quantities reported by the media that can help inform social distancing, the total number of cases $M$ and the active cases $M_A$ (each also scaled by $N_{\text{crit}}$) which are defined by

$$\dot{M} = \rho_S(I_{S_0} + I_{S_1} + I_{S_2}) + \rho_A(I_{A_0} + I_{A_1} + I_{A_2} + P_0 + P_1 + P_2) \tag{2.1a}$$

and

$$M_A = P_M + I_{S_M} + I_{A_M}. \tag{2.1b}$$

If the disease is in the exponential phase of spread then the doubling rate can be deduced from the cumulative case information, $M$, to yield

$$k_M = \frac{(\mathrm{d}M/\mathrm{d}t)}{M \ln 2}. \tag{2.2}$$

We assume that this can approximate the doubling rate at all times and is what locally drives social distancing. However, while the disease growth rate is important, it should be weighed against the number of active cases as well and therefore we propose a social distance transition function

$$\mu = \mu_{\max}\left(\frac{[k_M - k_c]_+}{[k_M - k_c]_+ + k_0 - k_c}\right)\left(\frac{[M_A - M_c]_+}{[M_A - M_c]_+ + M_0 - M_c}\right), \tag{2.3}$$

where $\mu_{\max}$ is the maximal rate of social distancing, $[\,\cdot\,]_+$ is defined such that,

$$[\,\cdot\,]_+ = \max(\,\cdot\,, 0).$$

Social distancing is not something that people want to do and the parameters $k_c$ and $M_c$ represent critical doubling rates and active case numbers, respectively, below which people will not social distance, which

is the role of the maximum function. These parameters can be thought of as policy parameters since implementing lock-downs, closing businesses and halting social gatherings will impact these values. The parameters $k_0$ and $M_0$ represent doubling rates and case numbers where social distancing reaches its half-maximum.

We assume that social relaxation, $v$, is proportional to the cost of social distancing, $c$, in dollars, which we model as

$$\dot{c} = \kappa N_{crit} ((S_2 + E_2) + (1 - \delta)(S_1 + E_1)). \tag{2.4}$$

The parameter $\kappa$ is the cost per person per day of being in social distancing class 2. Those in social distance state 1 effect their transmissibility by a factor $\delta$ and we assume this comes at a reciprocal burden cost of $(1 - \delta)$ relative to $\kappa$. For example, if $\delta = 1$ then the 1 and 2 states are both fully isolated and contribute an equal maximal cost. The parameter $N_{crit}$ appears in (2.4) because of the scaling on the populations. We only include susceptible and exposed classes in (2.4) because we assume there is a greater benefit to having transmitting classes ($P_i$, $I_i$) stay home. Arguably, exposed people who will soon become infectious should stay home too, but as they would test negative, they would think they are healthy and therefore we assume they contribute to the cost. For simplicity, we ignore the cost of recovered people social distancing which will be invalid if many people have recovered but policy prevents them from returning to their workplaces etc. As written, the cost accumulates with time. We could remedy this by including a decay factor $-\mu_c c$ in (2.5) but we assume that the time scale of recovery is much longer than that of the pandemic. Since $c$ does not factor in day-to-day economic costs in non-pandemic times, it is normalized so that zero cost represents the cost of society pre-pandemic. Similarly, then the maximum additional costs come from those isolating completely in social distance state 2.

Defining cost as we have in (2.4) attributes a single dollar amount, $\kappa$ to social distancing. This is a general opportunity cost which will vary from person to person and include direct economic costs in the form of people staying home from their jobs, but also indirect economic costs such as the psychological tolls of being isolated for a long period of time. As we have not stratified our model by demographics such as age and poverty level, we are not able to capture demographic effects on the cost. This generality in the model means that identifying an actual dollar amount per day, $\kappa$, is difficult. Instead, we define

$$c = \kappa N C$$

allowing us to eliminate $\kappa$ in (2.4) to yield,

$$\dot{C} = \frac{N_{crit}}{N} ((S_2 + E_2) + (1 - \delta)(S_1 + E_1)). \tag{2.5}$$

With this definition, $C$ is measured in days. Since $\kappa N$ represents the cost per day of every person in the population being full isolated then $C$ represents the equivalent cost in days of the entire province isolating. Interpreting a reasonable value of $\kappa$ will allow governments and policy makers to transform the cost into a daily total in dollars.

Defining the relaxation cost using (2.5), we propose $v$ be modelled by

$$v = v_{max} \frac{[C - C_c]_+}{[C - C_c]_+ + C_0 - C_c}, \tag{2.6}$$

where $v_{max}$ is the maximal rate at which social distancing can be relaxed, $C_0$ is the cost which triggers the half-maximal rate, and $C_c$ is the cost required to trigger social relaxation. $C_c$ is also a policy parameter as mental health promotion, economic stimulus, and wage subsidy programs can influence the cost people can endure before social relaxation.

To understand the true cost of the pandemic, we must balance the relaxation cost, $C$, with the overburden healthcare cost, $H$, which we define as

$$H = \sum_i \int_{t_{0_i}}^{t_{1_i}} M_A \, dt, \tag{2.7}$$

where $t_0$ is the time where active cases exceed $N_{crit}/2$ and $t_1$ is the time they return below $N_{crit}/2$. We choose this value as many provinces use this as an indicator of overload since by the time cases reach $N_{crit}$ resources are completely overwhelmed. An alternative definition of the healthcare cost could be to integrate over the whole duration of the pandemic. In this formulation, zero health cost could only

come from having no cases at all. Additionally, it means that a very small daily active caseload sustained over several years could be equivalent to or worse than an extreme overload of the system over a couple of weeks. By defining (2.7), we are assuming that the healthcare system has measures in place to manage caseloads below $N_{crit}/2$. Even small case numbers will contribute to death and chronic illness, but we assume that below $N_{crit}/2$ these are solely a function of the disease, while above the threshold, the impact on healthcare strain is likely to be a contributing factor.

The choice of integrating $H$ in (2.7) balances intensity of the outbreak along with duration. The sum allows for multiple outbreaks where the hospital resources are exceeded. The reason we measure active cases is that we assume all COVID-19 cases entering the hospital will be tested. Realistically a portion of the untested symptomatic cases will also impact the healthcare system and therefore this can be considered an underestimated cost. Having defined the overburden healthcare cost, we can then define the total cost as

$$C_T = \omega \frac{H}{H_\infty} + (1 - \omega)\frac{C}{C_\infty},$$ (2.8)

where $H_\infty$ is the overburden healthcare cost with no social distancing intervention ($\mu_{max} = 0$) and $C_\infty$ is the largest isolation cost allowable. We define $\omega$ as a weighting factor between the two cost contributions.

## 2.2. Parameter determination

We first consider a variant of the full model that does not include social distancing or testing (green-dash rectangle of figure 1) which is given by,

$$\left.\begin{aligned}
\dot{S} &= -\frac{N_{crit}}{N}(\beta_{SP}SP + \beta_{SI_S}SI_S + \beta_{SI_A}SI_A) \\
\dot{E} &= \frac{N_{crit}}{N}(\beta_{SP}SP + \beta_{SI_S}SI_S + \beta_{SI_A}SI_A) - \sigma E \\
\dot{P} &= \sigma E - \phi P \\
\dot{I}_S &= Q\phi P - \gamma_S I_S \\
\dot{I}_A &= (1 - Q)\phi P - \gamma_A I_A \\
\dot{R}_S &= \gamma_S I_S \\
\dot{R}_A &= \gamma_A I_A.
\end{aligned}\right\}$$ (2.9)

This reduced model represents the disease transmission dynamics prior to widespread knowledge of COVID-19. Following [16], we assume that people without symptoms are half as infectious as those with symptoms and therefore take $\alpha = 1/2$.

The disease free state is $[S, E, P, I_S, I_A, R_S, R_A] = [N/N_{crit}, 0, 0, 0, 0, 0, 0]$ and we identify the basic reproduction number $R_0$ as the non-zero eigenvalue of the next generation matrix produced from (2.9) (cf. [26,27]),

$$R_0 = \frac{\beta(\gamma_S\gamma_A + 2Q\phi\gamma_A + \phi\gamma_S(1 - Q))}{2\phi\gamma_S\gamma_A}.$$ (2.10)

Taking $R_0$ from measurements such as the studies in [16,17] which estimate $R_0 = 2.4$ we can rearrange (2.10) to determine a value for $\beta$.

We assume that this base transmission rate between the susceptible and symptomatic populations is the same as that between $S_0$ and $I_{S_0}$ in the social distancing model (A 1), i.e. $\beta_{S_0I_{S_0}} = \beta$. The parameters considered for the base model are presented in table 1 and we comment on some of the assumptions made.

As of 16 July 2020, the hospitalization rate of COVID-19 in Ontario, Canada was 12.3% and there were approximately 10 000 hospital beds available for people which together define $N_{crit} = 81\,301$. We choose $\mu_{max} = \nu_{max} = 1$ under the assumption that people generally need at least 1 day to change their routines. We arbitrarily assume that $C_0 = 2C_c$ and that $k_0 = 4k_c$ to help constrain the model. This means that the relaxation cost needed to initiate the half-maximum rate is twice as many days as the onset of social relaxation while the disease needs to double twice for the half-maximal social distancing rate to occur. We chose $C_c = 50$ based on Ontario imposing a stage-one lock-down in March 2020 that lasted almost 100 days coupled with the fact that it did not impact the entire province.

We predicted the values of $k_c$, $M_c$ and $\rho_A$ (recalling that $\rho_S = 4\rho_A$) by fitting our model to active and total case data from [15] between 10 March and 18 August 2020 inclusive. We used a nonlinear least-squares method for the fitting, the details of which are in appendix B. Using the $N_{\text{crit}}$ scale, we can convert the values of $M_c$ and $M_0$ from table 1 to 2090 and 4180 people, respectively. The values of $\mu_I$, $q_0$, $q_2$ and $q_I$ are arbitrarily chosen. However, as is seen in appendix B where a sensitivity analysis is performed, these parameters are not very influential on model results. The most influencing parameter is $\mu_I$. Considering that at the half-maximal rates, the social distancing rate is $\mu = 1/4$ and that a recent study from [28] suggested that up to 90% of Americans go into work sick then a further 90% reduction would yield $\mu_I \approx 0.025$ which is the same order of magnitude as the chosen value.

We took a median value for the symptomatic rate, $Q$, of 69% following a variety of studies (see [29–34]).

## 3. Results

We simulated (A 1) using Matlab 2020a with parameters in table 1. We took 10 March 2020 as the initial time with an initial condition that 0.02% of the population was infected with symptoms and placed the remaining 99.98% of the population in the susceptible class.

We demonstrate the results from data-fitting the parameters $k_c$, $M_c$ and $\rho_A$ in figure 2$a$,$b$. Comparing data with simulation, we observe a difference in the early peak-time of 4.7 days and a difference in peak active cases of 359 people. We extend our simulation beyond 18 August 2020 and compare with data up to 6 January 2021 in figure 2$c$,$d$. We see that the fit is good until around the end of September 2020. We discuss how to improve this fit in appendix B where we also discuss comparisons with social mobility data. We observe the impact of the disease on total cost (2.8) by simulating the full model (A 1) and varying the critical threshold at which people social distance ($M_c$) and the critical cost before social relaxation begins ($C_c$). We consider 1/4, 1/2, 2 and 4 times the base values given in table 1. The value for $k_c$ from data fitting is already quite extreme and we do not vary this. We plot heat maps for the total cost $C_T$ in figure 3 for different weights $\omega$ with the maps coloured relative to the maximal and minimal costs. We compute $H_\infty$ by simulating the model with parameters in table 1 and taking $\mu_{\max} = 0$. We take $C_\infty$ as the highest cost that emerges from all of the simulations. We note that $\omega = 0$ is just the relaxation cost $C$ given by (2.5) scaled by $C_\infty$ while $\omega = 1$ is just the overburden health cost $H$ given by (2.7) scaled by $H_\infty$. Reducing $M_c$ means that people require less active cases before triggering their social distance behaviour. If we denote this behaviour as vigilance then smaller values of $M_c$ lead to increased vigilance. Therefore, in figure 3, vigilance increases from bottom to top as $M_c$ decreases. $C_c$ increases from left to right, which corresponds to a higher tolerance for social distancing meaning that people delay their relaxation behaviour. We associate this to increased spending as people can absorb more cost.

When $\omega$ is small, corresponding to more weight being put on relaxation cost, figure 3 intuitively shows that increasing $C_c$ increases the total cost. This changes when $\omega$ approaches 1, where increasing $C_c$ can decrease total cost. This is also intuitive because when $\omega = 1$ there is no contribution of social distancing to the total cost, but the advantage that people are staying home and not getting sick. Therefore, encouraging that behaviour only leads to better outcomes. These two different behaviours suggest that there is a value of $\omega$ where both increasing and decreasing $C_c$ may lead to increased total costs. Indeed this phenomenon can be observed in figure 3$a$ where the optimal spending occurs at $M_c = M_c^*/4$ and $C_c = C_c^*$.

A very non-intuitive trend occurs in figure 3 which is that increasing vigilance (smaller $M_c$) increases total cost. The only exception to this is in figure 3$f$ when $C_c = 4C_c^*$ and the minimum total cost occurs when $M_c = M_c^*$. To understand this result, we focus on the case $\omega = 1$ where only the overburden health cost is considered. First consider $C_c = C_c^*/4$ and $M_c = M_c^*$. As vigilance increases, we expect that maximum active case load to decrease as people are social distancing with greater frequency. We see that this is indeed the case in figure 4$a$ as we change from $M_c^*$ to $M_c^*/2$. However, we also see that the duration of the epidemic straining healthcare resources is longer and the small decrease in peak is not enough to overcome this duration. We contrast this case to when $C_c = 4C_c^*$ and a minimum overburden healthcare cost is observed at $M_c = M_c^*$. The active case load is plotted in figure 4$b$ for $2M_c^*$, $M_c^*$, and $M_c^*/2$. Increasing vigilance from $2M_c^*$ to $M_c^*$ decreases the peak and increases the duration. However, unlike the case in figure 4$b$, the depression is significant enough to cause an overall decrease in total cost. However, as vigilance is increased further to $M_c^*/2$, the peak increases and duration decreases leading to an increase in cost. The non-intuitive result that higher vigilance leads to worse outcomes can be explained by isolation fatigue. A higher vigilance causes people to

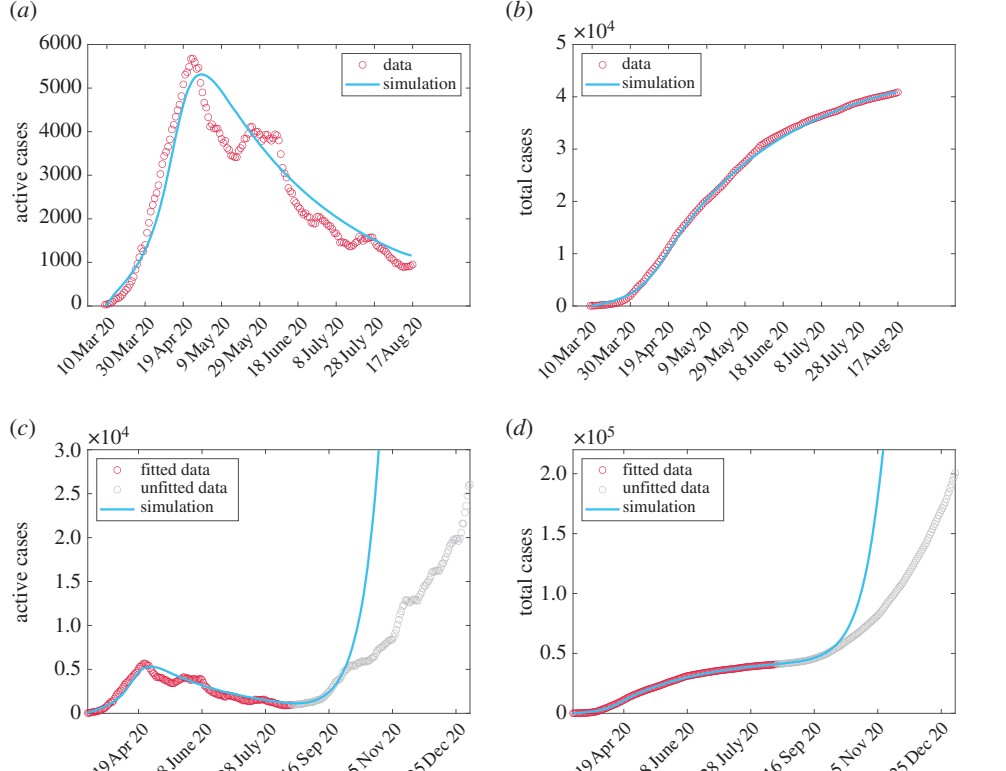

**Figure 2.** Comparison between simulation and data from [15]. Data are fitted from 10 March to 18 August 2020 (*a,b*) and then projected (*c,d*) with unfitted data from 19 August 2020 to 6 January 2021. Improvements to fit are discussed in appendix B.

enter isolation too early. Once they hit a certain cost threshold they relax back to their regular social habits and then cannot sustain further isolation when the second wave of the pandemic arrives.

We define $H_\infty$ in the case when no social distancing occurs and therefore the relative overburden healthcare cost $H/H_\infty$ tends to its maximum value of one as social distancing is relaxed. However, as demonstrated in figure 3, increasing vigilance can also lead to $H/H_\infty$ tending to 1. This suggests that for every spending level, $C_c$, there is an optimal social distancing $M_c$ to minimize the overburden health cost. Indeed, this is the case as demonstrated in figure 5 for the case $\omega = 1$ and $C_c = C_c^*/4$ where we report the critical number of active cases to isolate, $M_c$ as a percentage of the population $N$. The value $M_c^*$ is indicated by the black line and demonstrates that based on the data from Ontario, social distancing vigilance was initially too severe. This is a very important policy result since if isolation is too vigilant then the fatigue from isolation cost has a very negative impact long term. This result can easily be extended to other regions by using different parameters.

Plots of all of the active cases and costs for the scenarios in figure 3 can be found in the electronic supplementary material along with the populations of each isolation class to visualize the impact of social distancing and relaxation. The cumulative number of symptomatic-infected people for each scenario is plotted in figure 6. It can be seen that decreasing vigilance causes a more uniform accumulation of cases while increasing relaxation cost delays the peak of infection. In all cases, the same number of total people are infected as in the baseline case where testing occurs but no social distancing happens. This is expected as complacency and fatigue from NPIs eventually force the cost of social distancing to be too high for people to remain away from others. However, these delays can provide time for vaccination and other medical efforts to minimize the impact of the disease.

## 3.1. Multiple secondary waves

The model as derived only allows for one large secondary wave following the peak in Ontario around early May 2020. Since the relaxation rate, $\nu$, is solely a function of relaxation cost (2.5)

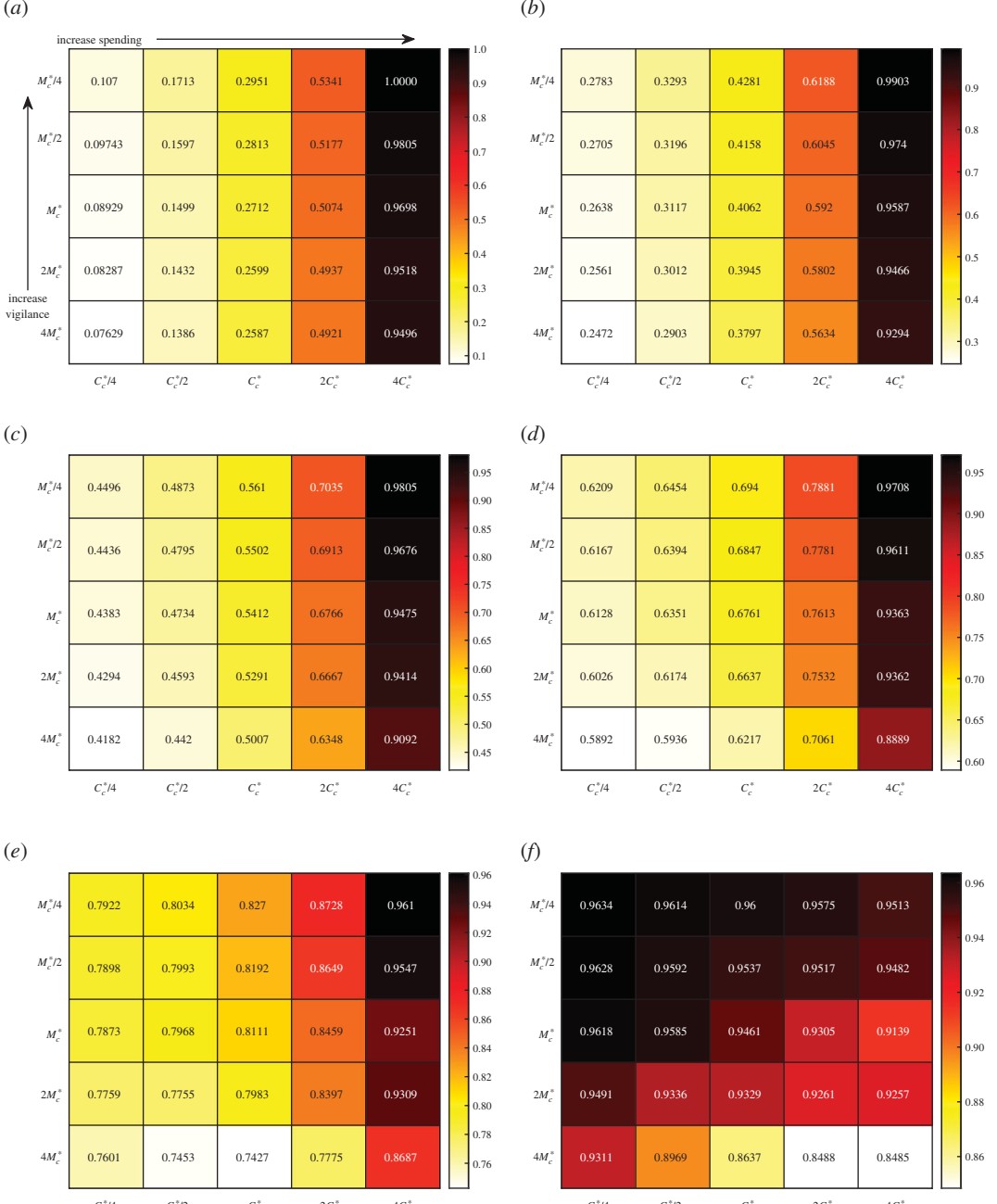

**Figure 3.** Total cost (2.8) from varying $M_c$ and $C_c$ in the model (A 1) with other parameters fixed from table 1 (excluding $C_0$ and $M_0$ which are appropriately updated). $M_c^*$ and $C_c^*$ refer to the base values in table 1. In these simulations, $H_\infty = 292.7$ and $C_\infty = 230$ days. Ascending the vertical axis corresponds to increased vigilance (lower required active cases before social distancing) while moving left-to-right on the horizontal axis corresponds to increased spending (longer tolerance before relaxing). (a) $\omega = 0$. $C_T = C/C_\infty$ with $C$ given by (2.5), (b) $\omega = 0.2$, (c) $\omega = 0.4$, (d) $\omega = 0.6$, (e) $\omega = 0.8$, (f) $\omega = 1$. $C_T = H/H_\infty$ with $H$ given by (2.7).

which is always increasing, isolation fatigue becomes too overwhelming that there is resistance for prolonged isolation. This model is likely to be appropriate for regions that have a strong aversion to social distancing. For other regions, it is likely that relaxation will be a function of cost and active cases as people will prioritize their health in a sustained outbreak and thus not want to relax if case numbers are sufficiently large. If we refer to the rate in (2.6) as $v_0$ then we propose modifying $v$ to

$$v = v_0[\eta M_c - M_A]_+, \tag{3.1}$$

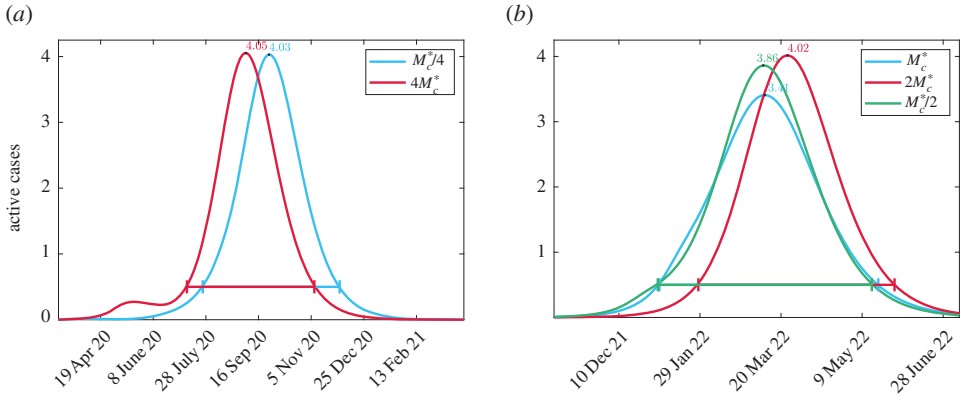

**Figure 4.** Comparison of active cases corresponding to figure 3f for different parameter values. (a) $\omega = 1$, $C_c = C_c^*/4$ and (b) $\omega = 1$, $C_c = 4C_c^*$.

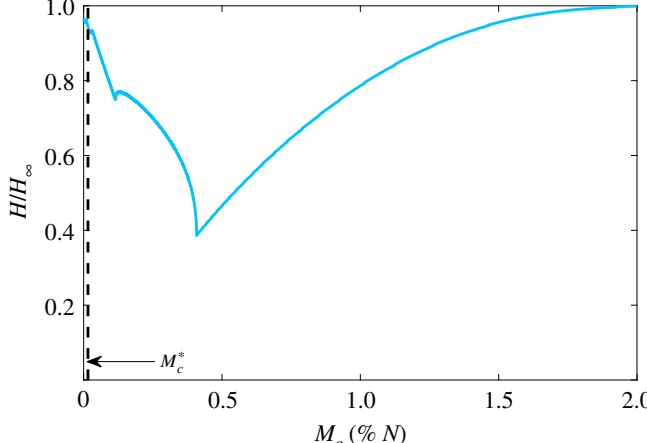

**Figure 5.** Overburden health cost, $H$, given by (2.7) (equivalent to total cost (2.8) when $\omega = 1$) when $C_c = C_c^*/4$ and $M_c$ is varied as a percentage of the total population $N$. The dashed black line indicates the value $M_c^*$ fitted from Ontario Public Health data [15].

where $\eta$ is a concern factor and is the number of critical cases $M_c$ that stops social relaxation regardless of cost. Implementing this change allows for secondary infection peaks as evidenced in figure 7 where we use parameters in table 1 and arbitrarily take $\eta = 1/2$ for figure 7a and $\eta = 1/5$ for figure 7b. It is important to note that these changes do not impact the initial peak fitted to data in figure 2 and only alter future projections. Furthermore, as of January 2021, Ontario is still in the midst of its first secondary peak. For these reasons, it is difficult to estimate $\eta$ as several peaks will need to have occurred.

We repeat the cost analysis as in figure 3 for the modified relaxation cost (3.1), however, we fix $M_c = M_c^*$, the value from table 1 and instead modify $\eta$. The results are presented in figure 8.

Figure 8 shows a different result compared with the case of figure 3, when (2.5) was used for the relaxation. In the latter case, there was a general trend upward in cost that had little difference between the value of $M_c$; however, in figure 8 there is a strong dependence in $\eta$. This is because the multi-secondary outbreaks caused by reducing relaxation with high active cases extends the duration of the epidemic which only increases the cumulative cost. The high impact of this is noticed as well with $C_\infty = 230$ days for figure 3, while $C_\infty = 614$ days for figure 8. $H_\infty$ remains the same in both cases since that is calculated with no social distancing at all (and therefore no social relaxation).

The introduction of the modified relaxation cost (3.1) has an impact on the health cost, as seen most dramatically in figure 8f where $\omega = 1$. For small values of $C_c$ when $\eta = 1/4$, there is no cost at all as the critical threshold is never reached. Non-intuitively, increasing spending (larger $C_c$) which provides incentive for people to stay home leads to worse health outcomes. The rationale for this is similar to what was observed in [8], where keeping people isolated for a longer duration increases their fatigue and resistance to staying isolated in future instances leading to large outbreaks. The impact of

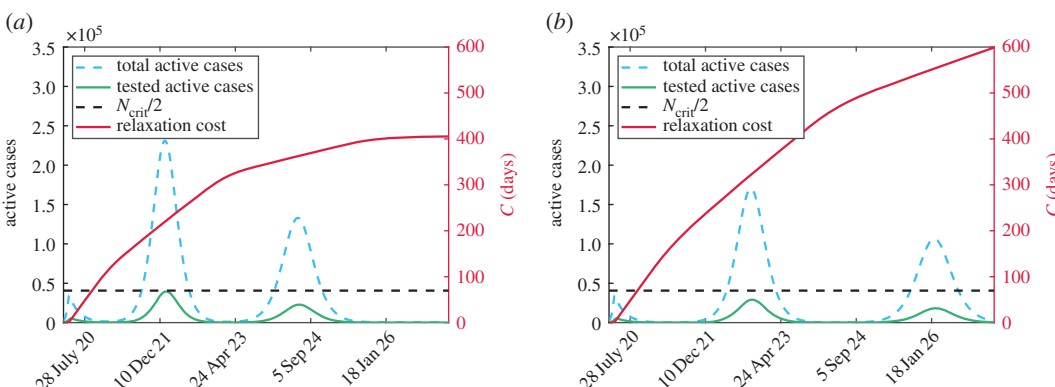

**Figure 6.** Total cumulative symptomatic cases (plotted as a percentage of the total population). The baseline case refers to no social distancing, i.e. $\mu_{max} = 0$. (a) $M_c^*/4$, (b) $M_c^*/2$, (c) $M_c^*$, (d) $2M_c^*$ and (e) $4M_c^*$.

**Figure 7.** Comparison of true active cases (dashed blue), tested active cases (solid green) and cost (solid red) for two values of $\eta$. The black-dashed line is $N_{crit}/2$ from which the health cost is measured. (a) $\eta = \frac{1}{2}$ and (b) $\eta = \frac{1}{5}$.

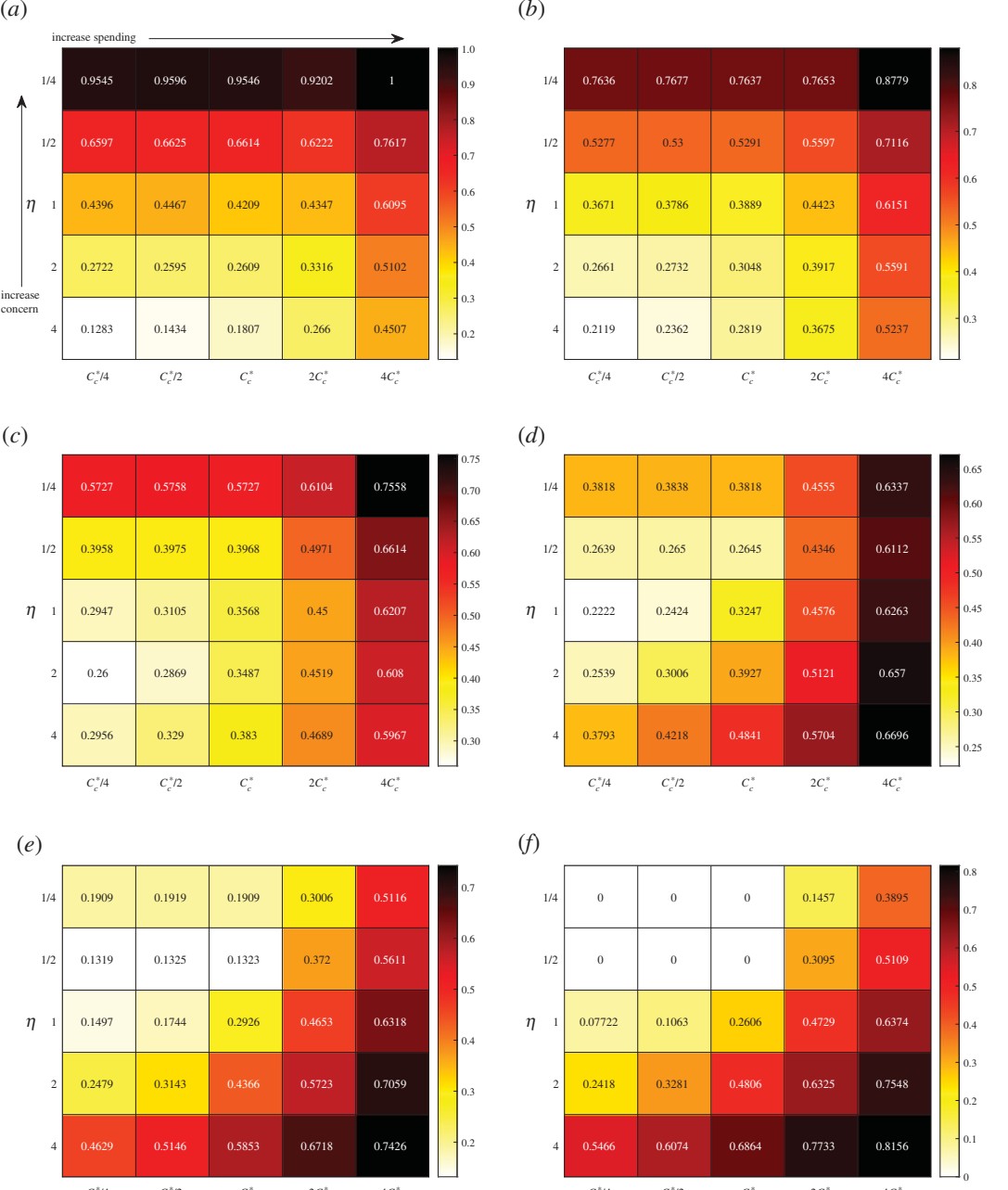

**Figure 8.** Total cost (2.8) using the relaxation cost (3.1) varying $\eta$ and $C_c$ in the model (A 1) with other parameters fixed from table 1 (excluding $C_0$ which is appropriately updated). $C_c^*$ refers to the base value in table 1. In these simulations, $H_\infty = 292.7$ and $C_\infty = 614$ days. Ascending the vertical axis corresponds to increased concern (lower required active cases before social relaxation stops) while moving left-to-right on the horizontal axis corresponds to increased spending (longer tolerance before relaxing). (a) $\omega = 0$. $C_T = C/C_\infty$ with $C$ given by (3.1), (b) $\omega = 0.2$, (c) $\omega = 0.4$, (d) $\omega = 0.6$, (e) $\omega = 0.8$. and (f) $\omega = 1$. $C_T = H/H_\infty$ with $H$ given by (2.7).

increased spending on active cases with $\eta$ fixed is demonstrated in figure 9. It is important to note that for a given spending $C_c$, the minimum total cost is not necessarily with the smallest value of $\eta$ (e.g. figure 8e when $C_c = C_c^*/4$). This is because there is a critical value of $\eta$ below which no additional healthcare savings occur but increasing expenses occur for relaxation. These results suggest a careful policy direction with more isolation periods of shorter duration.

The plots of cases and total infections for each of the scenarios in figure 8 can be found in the electronic supplementary material along with plots of each isolation class to demonstrate the social distancing and relaxing behaviour. We plot the cumulative symptomatic-infected proportions for each scenario in figure 10.

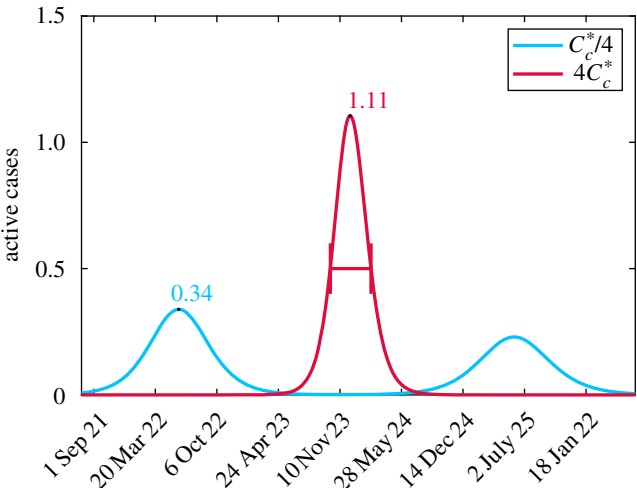

**Figure 9.** Comparison of the active cases when $\eta = 1/4$ for two values of $C_c$. When $C_c = 4C_c^*$ then there is a long period of time with no cases. However, a large outbreak forms since there is too much fatigue to isolate again when cases get large. Conversely, when $C_c = C_c^*/4$, a mild wave occurs early on because there is little incentive to isolate. However, since people were not isolating for very long, it is easier to endure further isolation when a second wave comes which leads to it also being mild. We note that when $C_c = C_c^*/4$ the critical threshold $N_{\mathrm{crit}}$ is never exceeded so there is no overburden healthcare cost.

Unlike the scenarios in figure 6 associated with the relaxation cost (2.5), basing relaxation on active cases as well can impact the terminal number of cumulative infections. Continually reducing $\eta$ decreases the total number of infected people which provides more evidence that increasing the relaxation cost threshold $C_c$ can cause more people to become infected. However, similar to figure 6, increasing $C_c$ leads to a longer delay before significant infection numbers occur.

# 4. Conclusion

We have presented a model for COVID-19 that allows for dynamic social distancing and relaxation based on the measured active cases and individual cost of isolating. The aim of this approach is that it more accurately reflects human behaviour and psychology unlike the modelling approach where behaviours are turned on and off at predetermined times. Understanding how people will react to a change in policy surrounding lock-downs or bans on social gatherings is essential in gauging the impact that COVID-19 and mitigation strategies will have on infections and mortality. Improving this modelling aspect can make sure that policies are put into place at the right time so people will react accordingly.

By modelling behaviour dynamically, we were able to produce non-intuitive results regarding the relative total cost of the disease, namely that increasing vigilance and relaxation cost does not necessarily lead to a decrease in total cost. This is because of the desire for people to socialize leading to isolation fatigue. We have demonstrated that in certain circumstances, however, the overburden healthcare cost can be eliminated entirely.

An advantage of the dynamic framework used in this model is that it is not restricted to Ontario nor is it even restricted to COVID-19. Changing the disease and behaviour parameters will allow this model to adapt to other scenarios. For COVID-19, policy makers would be advised to use data in the relatively early stages of a lock-down to fit behaviour parameters. Earlier time point data helps reduce the likelihood that the relaxation cost threshold has been exceeded so that the behaviour parameters are more accurate. Otherwise, $k_c^*$ and $M_c^*$ become stronger functions of the choice for $C_c^*$. As discussed in appendix B, there are also issues of assuming static parameters when the duration of time series is taken too long. The limited data and types of data available should discourage too much parameter fitting. Having determined the parameters to a given set of data, cost analysis using (2.8) can be done leading to results similar to figures 3 and 8. Understanding the influence of tangible actions such as forced closures, wage subsidies, etc. on parameters such as $M_c$, $C_c$ and $\eta$ probably requires surveys and other follow-up studies.

It is important to acknowledge that this model does not take into account vaccination or other pharmaceutical interventions. These have an important role in not only limiting the healthcare impact

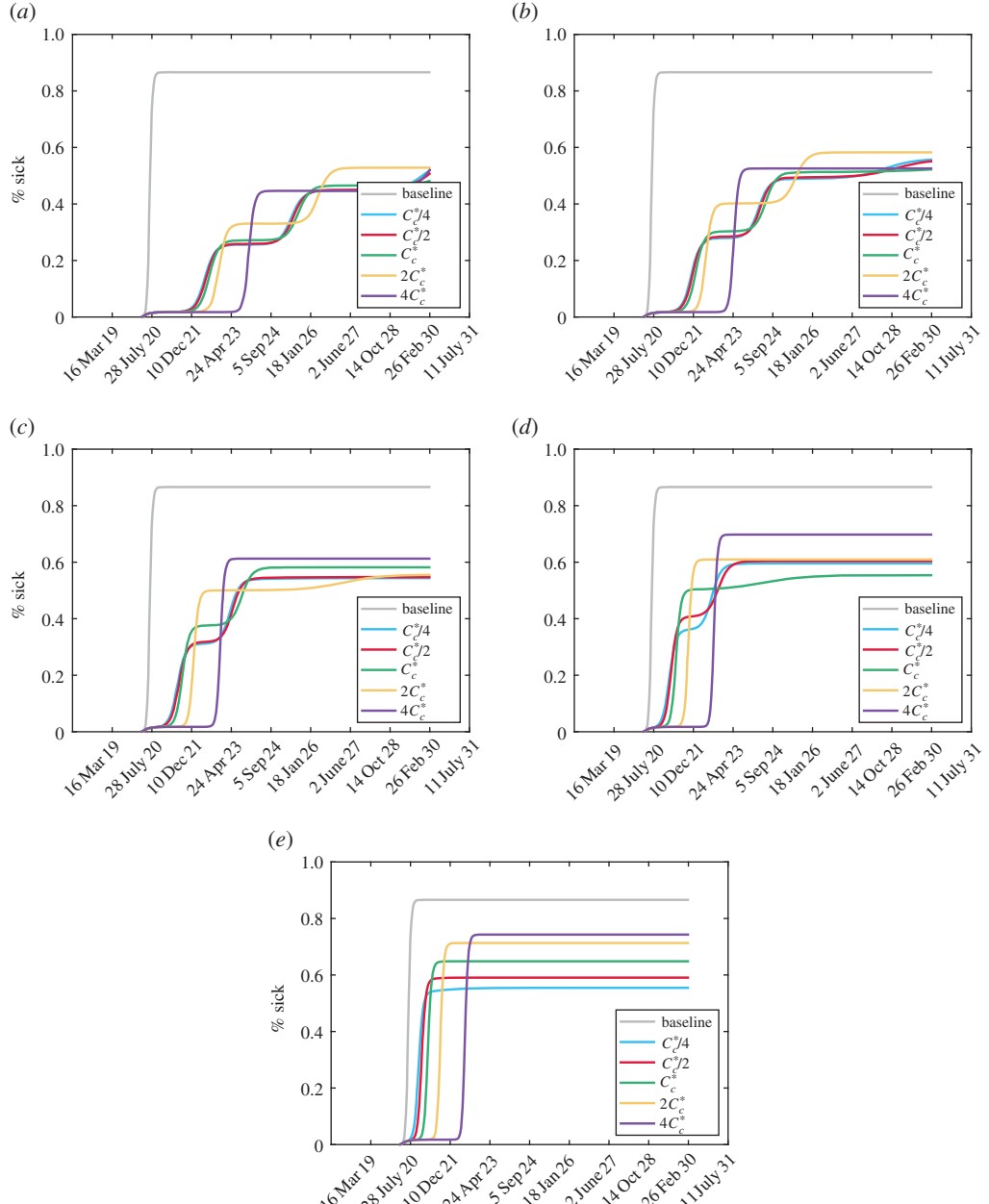

**Figure 10.** Total cumulative symptomatic cases (plotted as a percentage of the total population) with the modified cost function (3.1). The baseline case refers to no social distancing, i.e. $\mu_{max} = 0$. (a) $\eta = 1/4$, (b) $\eta = 1/2$, (c) $\eta = 1$, (d) $\eta = 2$ and (e) $\eta = 4$.

but also in outbreak peak time and duration. The introduction of the modified cost in (3.1) causes significant delays between peaks at increased spending. The duration of the pandemic can then be several years longer than when fatigue is strong resulting in a single large outbreak peak. This additional duration may be significantly longer than the time for an effective vaccine to be developed and deployed and this needs to be considered in future work. Another important consideration is that social distancing is not truly discrete in that people do not suddenly reduce their contacts. In reality, it is a spectrum with fluid contact rates and this needs to be further explored. Finally, taking the testing rates constant should be relaxed and reflect that both the testing capacity and willingness for individuals to test is a function of disease progression.

Data accessibility. The datasets and codes supporting this article have been uploaded as part of the electronic supplementary material. 'Data.zip' includes the data used 'Code.zip' includes all of the code used to run simulations and produce figures Inside 'Code.zip' is a README file with instructions on how to use codes and reproduce results.
Competing interests. We declare we have no competing interests.

Funding. I.R.M. acknowledges funding from an NSERC Discovery grant no. 2019-06337. J.M.H. acknowledges funding from an NSERC Discovery grant and Discovery Accelerator Supplement.
Acknowledgements. The authors are grateful to members from the Centre for Disease Modelling for feedback on the model. We thank Suzan Sardroodi for a thorough review of the manuscript.

# Appendix A. Differential equation model

The differential equation model visualized in figure 1 is given by

$$\dot{S}_0 = -F_{S_0} - \mu_{S_0 S_1} S_0 - \mu_{S_0 S_2} S_0 + \nu_{S_1 S_0} S_1 + \nu_{S_2 S_0} S_2$$

$$\dot{S}_1 = -F_{S_1} - \mu_{S_1 S_2} S_1 + \mu_{S_0 S_1} S_0 - \nu_{S_1 S_0} S_1 + \nu_{S_2 S_1} S_2$$

$$\dot{S}_2 = \mu_{S_0 S_2} S_0 + \mu_{S_1 S_2} S_1 - \nu_{S_2 S_0} S_2 - \nu_{S_2 S_1} S_2$$

$$\dot{E}_0 = F_{S_0} - \mu_{E_0 E_1} E_0 - \mu_{E_0 E_2} E_0 + \nu_{E_1 E_0} E_1 + \nu_{E_2 E_0} E_2 - \sigma E_0$$

$$\dot{E}_1 = F_{S_1} - \mu_{E_1 E_2} E_1 + \mu_{E_0 E_1} E_0 - \nu_{E_0 E_1} E_1 + \nu_{E_2 E_1} E_2 - \sigma E_1$$

$$\dot{E}_2 = \mu_{E_0 E_2} E_0 + \mu_{E_1 E_2} E_1 - \nu_{E_2 E_0} E_2 - \nu_{E_2 E_1} E_2 - \sigma E_2$$

$$\dot{P}_0 = \sigma E_0 - \mu_{P_0 P_1} P_0 - \mu_{P_0 P_2} P_0 + \nu_{P_1 P_0} P_1 + \nu_{P_2 P_0} P_2 - \phi P_0 - \rho_A P_0$$

$$\dot{P}_1 = \sigma E_1 - \mu_{P_1 P_2} P_1 + \mu_{P_0 P_2} P_0 - \nu_{P_1 P_0} P_1 + \nu_{P_2 P_1} P_2 - \phi P_1 - \rho_A P_1$$

$$\dot{P}_2 = \sigma E_2 + \mu_{P_0 P_2} P_0 + \mu_{P_1 P_2} P_1 - \nu_{P_2 P_0} P_2 - \nu_{P_2 P_1} P_2 - \phi P_2 - \rho_A P_2$$

$$\dot{P}_M = \rho_A (P_0 + P_1 + P_2) - \phi P_M$$

$$\dot{I}_{S_0} = Q\phi P_0 - \mu_{I_{S_0} I_{S_1}} I_{S_0} - \mu_{I_{S_0} I_{S_2}} I_{S_0} - \gamma_S I_{S_0} - \rho_S I_{S_0}$$

$$\dot{I}_{S_1} = Q\phi P_1 + \mu_{I_{S_0} I_{S_1}} I_{S_0} - \mu_{I_{S_1} I_{S_2}} I_{S_1} - \gamma_S I_{S_1} - \rho_S I_{S_1}$$

$$\dot{I}_{S_2} = Q\phi P_2 + \mu_{I_{S_0} I_{S_2}} I_{S_0} + \mu_{I_{S_1} I_{S_2}} I_{S_1} - \gamma_S I_{S_2} - \rho_S I_{S_2}$$

$$\dot{I}_{S_M} = \rho_S (I_{S_0} + I_{S_1} + I_{S_2}) + Q\phi P_M - \gamma_S I_{S_M}$$

$$\dot{I}_{A_0} = (1-Q)\phi P_0 - \mu_{I_{A_0} I_{A_1}} I_{A_0} - \mu_{I_{A_0} I_{A_2}} I_{A_0} + \nu_{I_{A_1} I_{A_0}} I_{A_1} + \nu_{I_{A_2} I_{A_0}} I_{A_2} - \gamma_A I_{A_0} - \rho_A I_{A_0}$$

$$\dot{I}_{A_1} = (1-Q)\phi P_1 - \mu_{I_{A_1} I_{A_2}} I_{A_1} + \mu_{I_{A_0} I_{A_1}} I_{A_0} - \nu_{I_{A_1} I_{A_0}} I_{A_1} + \nu_{I_{A_2} I_{A_1}} I_{A_2} - \gamma_A I_{A_1} - \rho_A I_{A_1}$$

$$\dot{I}_{A_2} = (1-Q)\phi P_2 + \mu_{I_{A_0} I_{A_2}} I_{A_0} + \mu_{I_{A_1} I_{A_2}} I_{A_1} - \nu_{I_{A_0} I_{A_2}} I_{A_2} - \nu_{I_{A_2} I_{A_1}} I_{A_2} - \gamma_A I_{A_2} - \rho_A I_{A_2}$$

$$\dot{I}_{A_M} = \rho_A (I_{A_0} + I_{A_1} + I_{A_2}) + (1-Q)\phi P_M - \gamma_A I_{A_M}$$

$$\dot{R}_{S_0} = \gamma_S I_{S_0}$$

$$\dot{R}_{S_1} = \gamma_S I_{S_1}$$

$$\dot{R}_{S_2} = \gamma_S I_{S_2}$$

$$\dot{R}_{S_M} = \gamma_S I_{S_M}$$

$$\dot{R}_{A_0} = \gamma_A I_{A_0}$$

$$\dot{R}_{A_1} = \gamma_A I_{A_1}$$

$$\dot{R}_{A_2} = \gamma_A I_{A_2}$$

$$\dot{R}_{A_M} = \gamma_A I_{A_M}$$

(A 1)

where $F_{s_i}$ is the force of infection,

$$F_{S_0} = \frac{N_{\text{crit}}}{N} (\beta_{S_0 P_0} S_0 P_0 + \beta_{S_0 I_{S_0}} S_0 I_{S_0} + \beta_{S_0 I_{A_0}} S_0 I_{A_0} + \beta_{S_0 P_1} S_0 P_1 + \beta_{S_0 I_{S_1}} S_0 I_{S_1} + \beta_{S_0 I_{A_1}} S_0 I_{A_1})$$

(A 2a)

$$F_{S_1} = \frac{N_{\text{crit}}}{N}(\beta_{S_1 P_0} S_1 P_0 + \beta_{S_1 I_{S_0}} S_1 I_{S_0} + \beta_{S_1 I_{A_0}} S_1 I_{A_0}$$
$$+ \beta_{S_1 P_1} S_1 P_1 + \beta_{S_1 I_{S_1}} S_1 I_{S_1} + \beta_{S_1 I_{A_1}} S_1 I_{A_1}), \tag{A 2b}$$

In deriving the model, we have normalized the population by $N_{\text{crit}}$ which represents the population that causes the healthcare system to be at capacity.

# Appendix B. Parameter fitting and model sensitivity

Many of the parameters associated with the natural progression of the disease are unknown as are several of the intervention parameters such as social distancing and testing. We use data from [15] for the 161 days between 10 March and 18 August 2020 on active and total cases to elucidate some key parameters. To do this, we solve our model (A 1) with the parameters in table 1 excluding $k_c$, $k_0$, $M_c$, $M_0$, $\rho_A$ and $\rho_S$ which we fit to the data. We run our model for 161 days using an initial condition that 0.02% of the population was initially infected with the remaining 99.98% being symptomatic. The actual number of people with COVID-19 is a matter of speculation and the arbitrary choice of the initial value will affect the fitting parameters, particularly the testing rates which are intimately linked. We do not partition any of the initial infected population into symptomatic or asymptomatic also due to the lack of clarity on true numbers. To help constrain the model, we take $k_0 = 4k_c$ so that the doubling rate has to double twice to trigger the half-maximal social distancing. We also take $M_0 = 2M_c$ so that the number of active cases need to double to trigger the half-maximal social distancing rate. Finally, since we assume that symptomatic people are more likely to get tested than asymptomatic people we take $\rho_S = 4\rho_A$. These constraints should not be too restricting, probably impacting the fitted values of the remaining free variables $k_c$, $M_c$ and $\rho_A$.

We use a nonlinear least-squares iterative procedure to identify the parameters. This leads to the values $k_c = 1/16.24$ d$^{-1}$, $M_c = 2.57 \times 10^{-2}$ and $\rho_A = 8.7 \times 10^{-3}$ d$^{-1}$ as in table 1 with a residual norm of $9.5 \times 10^{-3}$. A graphical representation of the fit between data and model is in figure 2.

## B.1. Extended fitting

The parameter fittings for figure 2 lead to excellent agreement beyond the fitting time window up to around 30 September 2020. From there, the model begins to overestimate the case load. Observing the data points, we note a series of intermittent plateaus are reached with a general overall increasing trend. One possible explanation for slower growth than we initially anticipated is that we have assumed a static parameter set. In reality, there will be dynamic changes to behaviour if strategies fail or cases continue to climb. In particular, people will probably have an increased sensitivity to the rate of change of cases more so than the actual number.

To improve the fit, we consider two values of $k_c$ and $k_0$, the one we have previously fitted in figure 2 and a second value associated with a panic in significant case increases. To fit the second set of values, we take the data from 6 September to 16 October 2020 and fit a new $k_c$ and $k_0$ with all other parameters fixed and taken from table 1. Using the same nonlinear least square method as for the first fit, this leads to $k_c = 0$ and $k_0 = 7.01 \times 10^{-2}$. We note that this is very reflective of a panic type behaviour whereby no growth rate will prevent social distancing and the rate required to reach the half-maximal growth is significantly reduced.

Defining a base set and panic set for $k_c$ and $k_0$ we simulate the model again. We assume that the base values hold until 6 September. From 6 September to 16 October, the panic values drive the model. Afterwards, we assume that behaviour alternates between base levels and panic levels, probably with fatigue so that panic cannot be sustained for very long. We plot a particular example of alternating behaviour in figure 11 where we take base values for 17 October to 10 November and then panic values for 10 days at which point we return to normalcy until 5 December 2020. We then take 10 more days of panic values before returning to normalcy for the remainder of the simulation.

We note that there are many ways to fit this data such as adjusting the time intervals for each behaviour or fitting a series of new parameters for each plateau region. Furthermore, the assumption of constant reporting rates probably does not reflect the day-to-day variability in testing, i.e. people opting for weekday tests over weekend tests. Therefore, we emphasize that the aim of figure 11 is to

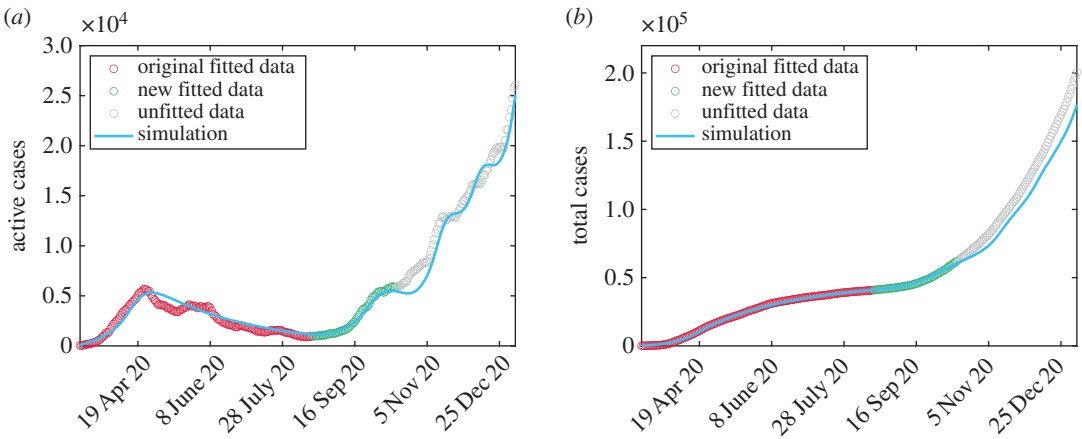

**Figure 11.** Modified data fitting to projected data taking account behavioural change due to large increases in active cases. (a) Active cases and (b) total cases.

demonstrate that a poor fit to data in the second wave as seen in figure 2 does not necessarily mean that the underlying model has failed, but that behavioural assumptions have probably changed.

### B.1.1. Sensitivity analysis

Having fit data to the model, we then performed a sensitivity analysis on the parameters $\rho_S$, $\rho_A$, $q_I$, $\mu_I$, $q_2$, $q_0$, $q$, $\gamma$, $\phi$, $\sigma$ and $\alpha$. We used table 1 for the fixed parameters and as mean values for the varying parameters, each distributed uniformly between maximum and minimum values. We used the Latin hypercube sampling technique with 10 000 iterations and a Spearman partial rank correlation coefficient (PRCC) to measure monotonicity. We tested the sensitivity to the cumulative infected (symptomatic and asymptomatic), susceptibles, peak time for the outbreak, and the value at the peak of the outbreak. We plot the results in figure 12 The most significant parameter is the recovery rate $\gamma$ (assumed the same for both classes) which seemingly has inverse behaviour to what is expected. That is, an increase in the recovery rate seems to cause more people to become sick. This is because the basic reproduction number is fixed at 2.4 and therefore changing $\gamma$ effectively changes the transmission $\beta$ making a higher recovery rate lead to a more transmissible disease. Aside from this, the most significant parameters unsurprisingly are the testing rates $\rho_S$ and $\rho_A$ as well as the symptomatic proportion $q$. This supports the importance of testing and social distancing. Interestingly, there is not much sensitivity to the peak time and value of the outbreak confirming the need for long-term planning regarding vaccination and hospital resource management.

We did not perform a sensitivity analysis on $C_c$ or $M_c$ as these are policy parameters. Their sensitivity is effectively measured by comparing costs in figures 3 and 8.

### B.1.2. Mobility data

Google has provided a dataset of community mobility created from users with the location history active on their mobile devices [35]. The dataset has six categories which are retail and recreation, grocery and pharmacy, parks, transit stations, workplaces, and residential. Most categories compare a percentage change in visitors to a baseline which is the median activity in a five-week period encompassing 3 January to 6 February 2020. The exception to this is residential category which measures the change in duration.

It is hard to directly compare mobility data and social distancing/relaxation. For example, there could be no change in activity at a store in terms of visitors, but changes in policy such as enforced physical distancing or limited capacity shopping. This means that the number of visitors to a location may not change even though their social distancing behaviour has. This is evidenced in the Google mobility data for grocery and pharmacy where relatively little change is observed despite significant policy changes including mandatory masks. The residential category offers limited insight as there are only 24 h in a day so duration variability cannot be that high.

We considered the retail and recreation category to be the most reflective of distancing activity as it is generally a non-essential activity. For a baseline in our model, we assumed that without COVID-19, there

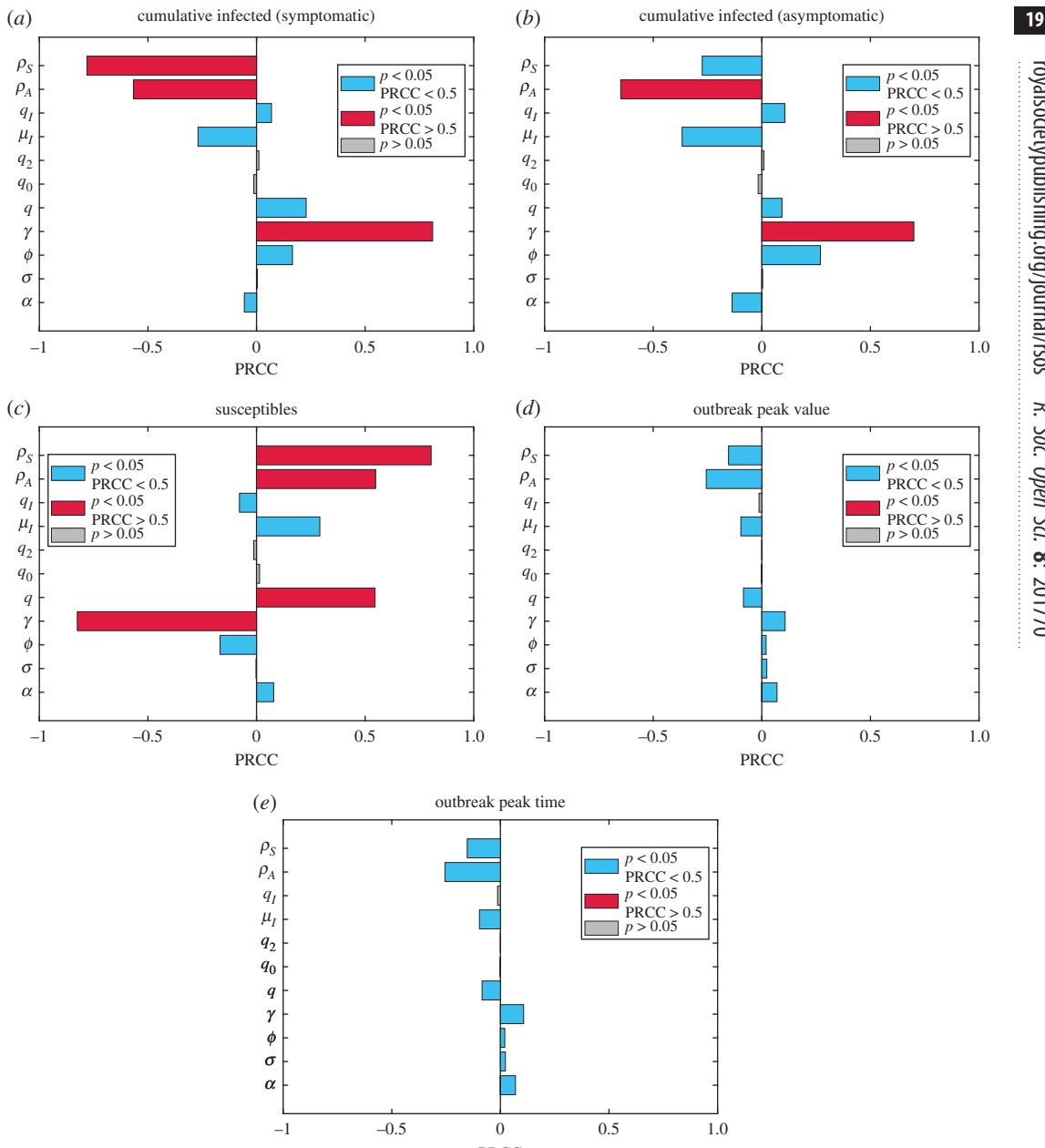

**Figure 12.** Sensitivity analysis of the model (A 1) using 10 000 iterations of a Latin hypercube sampling method with a Spearman partial rank correlation coefficient. Grey bars indicate a parameter with a $p$-value $p > 0.05$ dismissing their significance. If $p < 0.05$, the bar is blue unless it is strongly correlated (absolute PRCC greater than 0.5) in which case it is red.

would be no social distancing. If we define the social distancing population,

$$D = S_1 + S_2 + E_1 + E_2 + P_1 + P_2 + P_M + I_{S_1} + I_{S_2} + I_{S_M} + I_{A_1} + I_{A_2} + I_{A_M}, \tag{B 1}$$

i.e. it covers anyone who is isolating, then we consider the percentage change from baseline to be the fraction of the total population $D/N$ that is social distancing. We take this percentage to be negative since social distancing is a reduction in their activity. A comparison between our simulation and Google's community mobility data for Ontario from 10 March 2020 to 6 January 2021 is in figure 13. We emphasize the importance of qualitative trends. We note that our model predicts a similar uptake in isolation (about 55%). Our model reaches this level more gradually and then sustains it until a relaxation occurs whereas the data shows a steady increase to the relaxation point. Regional lock-downs

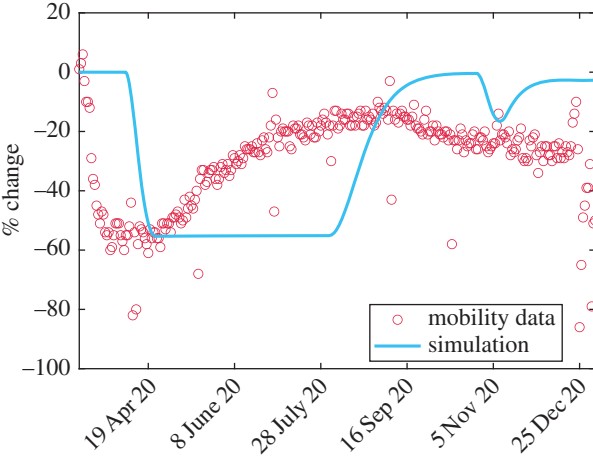

**Figure 13.** Comparison between simulation and mobility data from [35] for percentage change in retail activity from 10 March 2020 to 6 January 2021.

probably contributed to the very sudden drop in retail usage. Furthermore, as previously stated, an increase in retail activity does not mean it was done while decreasing isolation. We note that our model sees a second decrease in activity which also qualitatively fits with the retail mobility data.

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
