## [Peer Review File · Royal Society Open Science]

Review History

RSOS-201770.R0 (Original submission)

Review form: Reviewer 1 (Nicolas Rebuli)

Is the manuscript scientifically sound in its present form?

Yes

Are the interpretations and conclusions justified by the results?

Yes

Is the language acceptable?

Yes

Do you have any ethical concerns with this paper?

No

Have you any concerns about statistical analyses in this paper?

No

Recommendation?

Accept with minor revision (please list in comments)

Comments to the Author(s)

Hello,

Thank you for your submission, I think you have quite an interesting model and I look forward to seeing it published. I have only a few minor requests for you to elaborate on some subjects and I found a few grammatical problems so I suggest double-checking for them (see Appendix A).

Kind regards

Review form: Reviewer 2**Is the manuscript scientifically sound in its present form?**

Yes

Are the interpretations and conclusions justified by the results?

Yes

Is the language acceptable?

Yes

Do you have any ethical concerns with this paper?

No

Have you any concerns about statistical analyses in this paper?

No

Recommendation?

Accept with minor revision (please list in comments)

Comments to the Author(s)

Cost and social distancing dynamics in a mathematical model of COVID-19 with application to Ontario, Canada

This is a good theoretical modelling study of the covid-19 epidemic. The novelty arises from the consideration of the covid-fatigue aspect of social distancing, measured in the model as a cost, considered independently of the direct health costs. I think that this idea, of a kind of budget for lockdown/distancing patience among the public, is very interesting, and could be explored further.

One of the problems with this approach is that it is hard to compare the two kinds of costs on the same scale. The authors address this problem by generating plots under different scalings of the costs (ω) (e.g. Fig 7cd). Although this is not entirely satisfactory, it is a step in the right direction. Is it possible to include a graphical representation varying over more values of ω ?

I did not understand the choice in equation 2.6 around the healthcare costs. Specifically why the integral is not taken over the whole duration of the epidemic.

My main criticism is that the paper is quite long and the quality of writing is a little inconsistent. This detracts from the clarity of the new ideas that are being described. I recommend that the authors read over the manuscript carefully and try to tighten up the description and presentation of the results.

Minor comments:

1. abstract: "We observe that minimum costs are not always associated with increased spending and increased vigilance which is due to the desire for people to not distance and the fatigue they experience when they do." I don't understand this sentence.
2. abstract: last sentence: "extracting" should be "expanding" and should be "are presented"
3. abstract: "an increased" should be "an increase"
4. references: you need to use curly brackets to get the capitalization correct in many of the references here (assuming latex)
5. page 8 line 54 odd use of "exasperate"
6. use of significant figures is inconsistent - suggest a careful revision of all numerical values to give an appropriate level of precision

Decision letter (RSOS-201770.R0)

Dear Dr Moyles

The Editors assigned to your paper RSOS-201770 "Cost and social distancing dynamics in a mathematical model of COVID-19 with application to Ontario, Canada" have now received comments from reviewers and would like you to revise the paper in accordance with the reviewer comments and any comments from the Editors. Please note this decision does not guarantee eventual acceptance.

Please submit your revised manuscript and required files (see below) no later than 21 days from today's (ie 30-Nov-2020) date. Note: the ScholarOne system will 'lock' if submission of the revision is attempted 21 or more days after the deadline. If you do not think you will be able to meet this deadline please contact the editorial office immediately.

Please note article processing charges apply to papers accepted for publication in Royal Society Open Science (<https://royalsocietypublishing.org/rsos/charges>). Charges will also apply to papers transferred to the journal from other Royal Society Publishing journals, as well as papers submitted as part of our collaboration with the Royal Society of Chemistry

(<https://royalsocietypublishing.org/rsos/chemistry>). Fee waivers are available but must be requested when you submit your revision (<https://royalsocietypublishing.org/rsos/waivers>).

on behalf of Professor Joshua Ross (Associate Editor) and Mark Chaplain (Subject Editor)
openscience@royalsociety.org

Reviewer comments to Author:
Reviewer: 1

Comments to the Author(s)
Hello,

Thank you for your submission, I think you have quite an interesting model and I look forward to seeing it published. I have only a few minor requests for you to elaborate on some subjects and I found a few grammatical problems so I suggest double-checking for them.

Kind regards

Reviewer: 2

Comments to the Author(s)
Cost and social distancing dynamics in a mathematical model of COVID-19 with application to Ontario, Canada

This is a good theoretical modelling study of the covid-19 epidemic. The novelty arises from the consideration of the covid-fatigue aspect of social distancing, measured in the model as a cost, considered independently of the direct health costs. I think that this idea, of a kind of budget for lockdown/distancing patience among the public, is very interesting, and could be explored further.

One of the problems with this approach is that it is hard to compare the two kinds of costs on the same scale. The authors address this problem by generating plots under different scalings of the costs (ω) (e.g. Fig 7cd). Although this is not entirely satisfactory, it is a step in the right direction. Is it possible to include a graphical representation varying over more values of ω ?

I did not understand the choice in equation 2.6 around the healthcare costs. Specifically why the integral is not taken over the whole duration of the epidemic.

My main criticism is that the paper is quite long and the quality of writing is a little inconsistent. This detracts from the clarity of the new ideas that are being described. I recommend that the

authors read over the manuscript carefully and try to tighten up the description and presentation of the results.

Minor comments:

1. abstract: "We observe that minimum costs are not always associated with increased spending and increased vigilance which is due to the desire for people to not distance and the fatigue they experience when they do." I don't understand this sentence.
2. abstract: last sentence: "extracting" should be "expanding" and should be "are presented"
3. abstract: "an increased" should be "an increase"
4. references: you need to use curly brackets to get the capitalization correct in many of the references here (assuming latex)
5. page 8 line 54 odd use of "exasperate"
6. use of significant figures is inconsistent - suggest a careful revision of all numerical values to give an appropriate level of precision

===PREPARING YOUR MANUSCRIPT===

===PREPARING YOUR REVISION IN SCHOLARONE===

Author's Response to Decision Letter for (RSOS-201770.R0)

See Appendix B.

RSOS-201770.R1 (Revision)

Review form: Reviewer 1 (Nicolas Rebuli)

Is the manuscript scientifically sound in its present form?

Yes

Are the interpretations and conclusions justified by the results?

Yes

Is the language acceptable?

Yes

Do you have any ethical concerns with this paper?

No

Have you any concerns about statistical analyses in this paper?

No

Recommendation?

Accept with minor revision (please list in comments)

Comments to the Author(s)

(1) Fig 1 appears to have a red and green line on top of one another. Consider separating these.

(2) Every graph with "t (days since March 10)" on the x-axis should be modified to have the date on the x-axis.

(3) Pg 14 point iv. In Australia, an individual who has been identified through contact tracing as somebody who is at risk of having contracted the virus is forced to complete a test. This is virtually the only way an asymptomatic individual case can possibly be identified. Although I can't comment on the capabilities and policies of the Ontario contact tracers, I find it hard to believe that 'curiosity' is a driving factor behind ascertainment of asymptomatic cases.

(4) Section B (i). The author seems to be suggesting that a series of "plateaus" in the observed data are reflective of an apparent dynamical feed-back loop between incidence and social distancing. A quick look at the daily incidence counts for Ontario reveals a distinct 7-day periodicity from November 1 onwards. Investigation of the "Episode Date" reveals that the best estimate of disease onset is a function of the specimen collection date. Note also that this periodicity is not present in hospitalisations or deaths.

I suspect that the "plateaus" are actually a feature of how the data was collected.

<https://www.publichealthontario.ca/en/data-and-analysis/infectious-disease/covid-19-data-surveillance/covid-19-data-tool?tab=trends>

Aside from this, I'm not sure what the value of this section is.

(5) Figure 13. I was mainly interested in the social distancing dynamics of your model considering that it is such a significant part of the model. In particular, I believe an important step in your model validation should be to assess how realistic the social distancing dynamics are and I see comparison to mobility data as a useful way of doing this.

I note that social distancing should begin sooner and be more gradual. However, your incidence is well-calibrated within the fitting window.

Review form: Reviewer 2

Is the manuscript scientifically sound in its present form?

Yes

Are the interpretations and conclusions justified by the results?

Yes

Is the language acceptable?

Yes

Do you have any ethical concerns with this paper?

No

Have you any concerns about statistical analyses in this paper?

No

Recommendation?

Accept as is

Comments to the Author(s)

No further comments

Decision letter (RSOS-201770.R1)

Dear Dr Moyles

On behalf of the Editors, we are pleased to inform you that your Manuscript RSOS-201770.R1 "Cost and social distancing dynamics in a mathematical model of COVID-19 with application to Ontario, Canada" has been accepted for publication in Royal Society Open Science subject to minor revision in accordance with the referees' reports. Please find the referees' comments along with any feedback from the Editors below my signature.

Please submit your revised manuscript and required files (see below) no later than 7 days from today's (ie 08-Feb-2021) date. Note: the ScholarOne system will 'lock' if submission of the revision is attempted 7 or more days after the deadline. If you do not think you will be able to meet this deadline please contact the editorial office immediately.

on behalf of Professor Joshua Ross (Associate Editor) and Mark Chaplain (Subject Editor)
openscience@royalsociety.org

Reviewer comments to Author:

Reviewer: 1
Comments to the Author(s)

- (1) Fig 1 appears to have a red and green line on top of one another. Consider separating these.
- (2) Every graph with "t (days since March 10)" on the x-axis should be modified to have the date on the x-axis.
- (3) Pg 14 point iv. In Australia, an individual who has been identified through contact tracing as somebody who is at risk of having contracted the virus is forced to complete a test. This is virtually the only way an asymptomatic individual case can possibly be identified. Although I can't comment on the capabilities and policies of the Ontario contact tracers, I find it hard to believe that 'curiosity' is a driving factor behind ascertainment of asymptomatic cases.
- (4) Section B (i). The author seems to be suggesting that a series of "plateaus" in the observed data are reflective of an apparent dynamical feed-back loop between incidence and social distancing. A quick look at the daily incidence counts for Ontario reveals a distinct 7-day periodicity from November 1 onwards. Investigation of the "Episode Date" reveals that the best estimate of disease onset is a function of the specimen collection date. Note also that this periodicity is not present in hospitalisations or deaths.

I suspect that the "plateaus" are actually a feature of how the data was collected.

<https://www.publichealthontario.ca/en/data-and-analysis/infectious-disease/covid-19-data-surveillance/covid-19-data-tool?tab=trends>

Aside from this, I'm not sure what the value of this section is.

(5) Figure 13. I was mainly interested in the social distancing dynamics of your model considering that it is such a significant part of the model. In particular, I believe an important step in your model validation should be to assess how realistic the social distancing dynamics are and I see comparison to mobility data as a useful way of doing this.

I note that social distancing should begin sooner and be more gradual. However, your incidence is well-calibrated within the fitting window.

Reviewer: 2

Comments to the Author(s)

No further comments

===PREPARING YOUR MANUSCRIPT===

Your revised paper should include the changes requested by the referees and Editors of your manuscript. You should provide two versions of this manuscript and both versions must be provided in an editable format:
 one version identifying all the changes that have been made (for instance, in coloured highlight, in bold text, or tracked changes);
 a 'clean' version of the new manuscript that incorporates the changes made, but does not highlight them. This version will be used for typesetting.
 Please ensure that any equations included in the paper are editable text and not embedded images.

===PREPARING YOUR REVISION IN SCHOLARONE===

To revise your manuscript, log into <https://mc.manuscriptcentral.com/rsos> and enter your Author Centre - this may be accessed by clicking on "Author" in the dark toolbar at the top of the

page (just below the journal name). You will find your manuscript listed under "Manuscripts with Decisions". Under "Actions", click on "Create a Revision".

<https://royalsociety.org/journals/authors/author-guidelines/#supplementary-material> to include a suitable title and informative caption. An example of appropriate titling and captioning may be found at https://figshare.com/articles/Table_S2_from_Is_there_a_trade-off_between_peak_performance_and_performance_breadth_across_temperatures_for_aerobic_sc_ope_in_teleost_fishes_/3843624.

Author's Response to Decision Letter for (RSOS-201770.R1)

See Appendix C.

Decision letter (RSOS-201770.R2)

Dear Dr Moyles,

It is a pleasure to accept your manuscript entitled "Cost and social distancing dynamics in a mathematical model of COVID-19 with application to Ontario, Canada" in its current form for publication in Royal Society Open Science.

COVID-19 rapid publication process:

We are taking steps to expedite the publication of research relevant to the pandemic. If you wish, you can opt to have your paper published as soon as it is ready, rather than waiting for it to be published the scheduled Wednesday.

This means your paper will not be included in the weekly media round-up which the Society sends to journalists ahead of publication. However, it will still appear in the COVID-19 Publishing Collection which journalists will be directed to each week (<https://royalsocietypublishing.org/topic/special-collections/novel-coronavirus-outbreak>).

If you wish to have your paper considered for immediate publication, or to discuss further, please notify openscience_proofs@royalsociety.org and press@royalsociety.org when you respond to this email.

on behalf of Professor Joshua Ross (Associate Editor) and Mark Chaplain (Subject Editor)
openscience@royalsociety.org

Appendix A

Summary

The author constructs an SEIR ODE model of SARS-CoV-2 transmission in which the disease transmission parameters are scaled by social distancing behavior. The propensity to increase social distancing is a function of the exponential growth rate and the total number of detected cases. The propensity to stop socially distancing is a function of the integral of the total number of non-infectious individuals who are socially distancing and the total number of detected cases. Thresholds are placed on all these values (below which they have a relatively small effect) which are to be interpreted as policy levers. For instance, the authors interpret the threshold holding back social relaxation as representative of the cumulative effect of mental health promotion, economic stimulus, and wage subsidy programs.

The authors quantify the total “cost” of the epidemic as a weighted sum of a cost to the economy and a cost to the health system. The economic cost is proportional to the integral of the total number of non-infective people socially distancing. The health care cost is proportional to the integral of the total number of detected cases over all subsets of time where the health care system is half-way towards capacity (meant to represent the risk of an increase in the mortality rate).

The cost functions lead to an unintuitive outcome whereby the lowest total cost is generally observed when a low threshold is placed on social relaxation. The authors claim that long lock downs result in high fatigue and resistance to further lock downs, causing the population to approach subsequent outbreaks with indifference. Hence, the prevailing policy directive is for governing bodies to minimize the duration of lock downs.

Comments

1. This is an interesting way to balance transmission and social distancing. However, I noticed that you haven't done any validation of your social distancing dynamics against existing data. Can I suggest adding something where you compare the total number of people socially distancing in your model to some of the existing mobility data? My suggestion would be to compare against the Google “time spent at home” series which shows the relative change in the amount of time people are spending at home.
2. Given that your model is suggesting that governments minimize the duration of lockdowns, I was wondering what social relaxation actually looks like in your model. Particularly in relation to daily incidence.
3. I found Figure 1 a little confusing. I understand you've made an effort to present a minimal description of a very complex model so I'm just going to list a few of the things I was initially confused about when I looked at this graph.
 - a. It's not clear what δ is referring to.
 - b. If shading is proportional to transmission potential, then should IA1 be a lighter shade?
 - c. Should the rate from an S to an E be $\beta * f(x(t))$?
 - d. Which direction of arrow do ν and μ belong to? Do transitions between the 0 and 1 social distancing classes use the same rates?
 - e. For the P0, P1, P2 transitions into PM, can you change the arrows coming out of P1 and P2 into lines? Similar for the IA and IS classes.
4. I was a bit confused by the definition of q : *Whenever there are multiple paths for a class, a parameter $q_i \in [0,1]$ is used to partition them into each final compartment.* Based on Table

1, I assume you're referring to q_0 , q_1 , q_2 here? In that case I think this definition could be improved.

I think it's also important to distinguish between q_i and q . They both fit into the above definition but there's no subscript on q which is a bit confusing.

5. In regards to (P7L12): *Asymptomatic people are likely to only seek a test out of curiosity or if they have been in contact with someone who has tested positive*. I think the main reason for detecting an asymptomatic is through contact tracing or through a dedicated "testing blitz", as we call them in Australia.
6. I have a couple of comments in regards to C. Given that this is intended as a policy piece and that C plays a large part in the results, I think it's important to have a better understanding of C.
 - a. The definition of C (P8), *This may be [an] economic cost in the form of people staying home from their jobs, but also the psychological cost of being isolated for a long period of time*. I found this to be a little ambiguous. Is C the economic toll, the mental health toll or both?
 - b. Is C the cost to the whole economy or the individual? As a result, what assumptions are you making about the population and their decision making processes? Could you be over representing some demographics while underrepresenting others?
 - c. You may like to consider that C is similar to the concept of an *opportunity cost* from economics, which quantifies the difference between a realized outcome and what would have been the optimal outcome.
7. In regards to the definition of H (P8), you say t_0 starts from when MA begins to *strain* the healthcare system and then suggest $MA > N_{crit}$ as the system being *exceeded* and $MA > N_{crit}/2$ is the start of the system being *burdened*. So it wasn't immediately clear to me if t_0 is when $MA=N_{crit}$ or $N_{crit}/2$.
8. Looking at Figure 6 from the appendix, I can't see what's happening in your model. Could you consider removing the baseline and rescaling y_{max} on the left axis so we can see all outbreaks from the blue line?

Minor comments

Just a few minor comments on where I think the authors may have made a mistake or I found something difficult to read. This is not a complete list.

1. Abstract: We demonstrate that an **increased** in the number of lockdowns – should this say increase?
2. P4 L19: Therefore a **dynamics** social distancing model – should this say dynamical?
3. P4 L57: listed in Table 1 where we – space.
4. P5 L11: M which represents the mitigation **on** spread – should that say of?
5. P6 L31: μ_{max} defined twice in table – I think you missed ν_{max} .
6. P7 L11: we assume that symptomatic people are more likely to seek out a test as they **would** have symptoms. – they do have symptoms.
7. P7 L12: Despite the fact that testing numbers fluctuate with the progression of the disease we take the testing rates to be constant which makes them an **effective average testing rate**. – I just thought this wording was a bit strange.
8. P8 L14: This may **be economic** cost – should this say *an* economic cost?
9. P11: I found this colour palette confusing, suggest something like autumn or winter.
10. P11 L50: The overall trend, particularly with even weighting, is unsurprising, namely that increased vigilance (larger M_c) and increased spending (larger C_c) contribute to a larger cost. – I found this sentence a little confusing.
11. P12 L18: $C^*_C/2$ – incorrect capitalisation of subscript.
12. P16 L23: $\eta=2$ – missed the 1.

13. Please include a legend on Figures 6 and 9 from the main text and all the figures in the appendix.

Appendix B

Response to review comments for

Cost and social distancing dynamics in a mathematical model of COVID-19 with application to Ontario, Canada

Authors: I. R. Moyles, J. M. Heffernan, J. D. Kong

submitted to Royal Society Open Science, RSOS-201770

We are including a revised version of their article which addresses the concerns of the reviewers.

The broader changes to the work have been a rewrite of the results to better clarify the interpretation of the heatmaps and a more thorough derivation and explanation of the social cost C . We highlighted one of the impacts of the non-intuitive result about social distancing by including a new Figure (Figure 5, page 13) which shows the optimal social distancing for minimal health cost (under a given set of parameters). The data from Ontario indicates that initially the social distancing level was probably too high, and fatigue set in early.

We added two new sections to appendix B. Firstly, as time has passed since the submission of the first draft, we have updated the data from Public Health Ontario. We see that our fit is good up until around the end of September at which point, we begin to overestimate cases. While this does not detract from the story we tell, we wanted to add some discussion to show that a small modification can realign the agreement between data and simulation. We do this in subsection (i) Extended Fitting in Appendix B. Secondly, it was suggested by one of the reviewers that we discuss mobility data. This data is very qualitative, so we did not use it to infer our parameters, but we discuss some of the qualitative features that we capture. This is now in section (b) Mobility Data of Appendix B.

We outline the specific major concerns of each reviewer below. We have addressed the minor comments, but we do not reference them directly. We thank both reviewers for the thorough consideration of our manuscript which has led to an overall improvement in the quality and presentation.

Throughout the manuscript we have highlighted text in **red** when significant changes have been made.

Reviewer 1:

1. This is an interesting way to balance transmission and social distancing. However, I noticed that you haven't done any validation of your social distancing dynamics against

existing data. Can I suggest adding something where you compare the total number of people socially distancing in your model to some of the existing mobility data? My suggestion would be to compare against the Google “time spent at home” series which shows the relative change in the amount of time people are spending at home.

The reviewer brings up an excellent point and a great reference for mobility data. One of the issues with mobility data is that it is not quite the same thing as social distancing. For instance, the data from Google shows almost no change in grocery and pharmacy visits throughout the pandemic. This makes sense since both are very essential, but it does not mean that the interaction behaviour did not change. Many stores now place limits on shoppers, provide barriers for their checkout staff, and have in-store traffic monitoring. Furthermore, a province-wide mask order was put into effect in Ontario at the start of summer 2020. For these reasons we are hesitant to use mobility data for quantitative comparisons.

However, we have added a section in Appendix B on mobility data (page 24) where we do a qualitative comparison. We did not use the residential time as the reviewer suggested. Google acknowledges that this data is hard to interpret due to the limit that there are only 24 hours in a day. Therefore, large variations in time spent at home are not possible. Instead, we isolated the retail and recreation data set from Google as this capture a lot of non-essential activity. We defined a percentage behavioural change in our model as the fraction of the population social distancing at any level since we assume that before COVID-19 this was not happening. The comparison from our simulation and data (Figure 13, page 24) do not match pointwise, but we do capture some similar phenomena. For example, we see a similar bottom limit in reduction of activity and a similar relaxation. Our transition is much sharper than what appears in the data, but we associate this to many of the reasons above, i.e., people may have maintained their social distancing behaviour, but increased their shopping activity. Furthermore, the data captures sharp discrete events such as retail lockdowns which are not present in our model. Nevertheless, we believe our qualitative agreement is encouraging and we thank the reviewer for the suggestion.

2. Given that your model is suggesting that governments minimize the duration of lockdowns, I was wondering what social relaxation actually looks like in your model. Particularly in relation to daily incidence.

The social distancing and relaxation are what is controlled by the parameters M_c (distancing) and C_c (relaxing). However, the reviewer raises a good point that we showed the impact on active cases by varying those parameters, but not the mobility within the susceptible group to see where people are isolating/relaxing to. We have thus now included the social distancing populations as part of the supplementary figures. We have equivalent figures to match each of the scenarios analyzed for the active cases. These figures show large and small peaks of social distancing activity as the disease progresses.

3. I found Figure 1 a little confusing. I understand you’ve made an effort to present a

minimal description of a very complex model so I'm just going to list a few of the things I was initially confused about when I looked at this graph.

- (a) It's not clear what δ is referring to.

δ originally referred to the parameter that decreased the transmission from social distancing. We have removed the parameter for the diagram since the colour fading already captured this effect.

- (b) If shading is proportional to transmission potential, then should IA1 be a lighter shade?

Thank you for the suggestion to fade the colour on I_A , we have done this. In the graphic description, we have indicated that the quantification of colour is represented by α and δ in the full model of the appendix.

- (c) Should the rate from an S to an E be $\beta f(x(t))$?

Yes, thank you. We have included the force of infection F as suggested.

- (d) Which direction of arrow do ν and μ belong to? Do transitions between the 0 and 1 social distancing classes use the same rates?

To help understand which arrow belongs to μ and ν we have done two things. Firstly, we have pushed the parameters tighter to each box so that they are closer to the arrow they represent. Also, we have matched the colours of the arrowhead and their corresponding parameter for the arrow that shows the parameters. Each transition does not necessarily use the same rate, we have made explicit in the text that each arrow would in theory have its own parameter and we just put a representative parameter to de-clutter the graphic. We emphasize in the figure caption that the main text clarifies how parameter rates are chosen. Arrows have been changed to lines as requested. We enlarged the graphic to also help the reader better see all of the transitions.

- (e) For the P0, P1, P2 transitions into PM, can you change the arrows coming out of P1 and P2 into lines? Similar for the IA and IS classes.

Great point, we have fixed this.

4. I was a bit confused by the definition of q : Whenever there are multiple paths for a class, a parameter $q_i \in [0, 1]$ is used to partition them into each final compartment. Based on Table 1, I assume you're referring to q_0, q_1, q_2 here? In that case I think this definition could be improved. I think it's also important to distinguish between q_i and q . They both fit into the above definition but there's no subscript on q which is a bit confusing.

The reviewer is correct to point out the confusion. Our initial attempt was to condense the definition, but as the reviewer points out there are not that many q parameters and so the definition is both confusing and unnecessary. We have remedied this in two ways. Firstly, to avoid the ambiguity with q with and without subscripts we have saved lower-case q for all the social distancing parameters, $q_0, q_2,$ and q_I and now

for the symptomatic-asymptomatic proportion use upper-case Q . Secondly, since each parameter was explained where necessary (and also in Table 1) we have removed the definition with the general subscript i .

5. In regards to (P7L12): Asymptomatic people are likely to only seek a test out of curiosity or if they have been in contact with someone who has tested positive. I think the main reason for detecting an asymptomatic is through contact tracing or through a dedicated “testing blitz”, as we call them in Australia.

Indeed, these are great points. We have clarified this in the manuscript (page 6).

6. I have a couple of comments in regards to C . Given that this is intended as a policy piece and that C plays a large part in the results, I think it’s important to have a better understanding of C .
 - (a) The definition of C (P8), This may be [an] economic cost in the form of people staying home from their jobs, but also the psychological cost of being isolated for a long period of time. I found this to be a little ambiguous. Is C the economic toll, the mental health toll or both?
 - (b) Is C the cost to the whole economy or the individual? As a result, what assumptions are you making about the population and their decision making processes? Could you be over representing some demographics while underrepresenting others?
 - (c) You may like to consider that C is similar to the concept of an opportunity cost from economics, which quantifies the difference between a realized outcome and what would have been the optimal outcome.

Our initial philosophy with the definition of cost was to begin with a reduced-dimensional formulation in a similar way to how we defined the populations. However, as the reviewer points out this leaves the interpretation quite vague. Furthermore, as days is a very non-standard unit of cost, we decided it would benefit with an improved derivation. We have rewritten the text on page 7 where we define cost. We start with a new dimensional cost, c , in dollars which is the sum of people social distancing in class 1 or 2. An important parameter in this initial formulation is κ measured in dollars/person/day which is the dollar cost of being in full isolation (class 2). As defined in the original presentation we assume that since class 1 has a reduction in isolation activity of δ they have a modified cost reduction $(1-\delta)$ compared to κ . An alternative derivation could consider separate κ for each group.

Since the new cost parameter κ is in dollars per person, we hope with our new formulation that it is now clear that the cost is a cumulative cost of individual isolation. The reviewer asks a pertinent question about what this cost actually is. This depends on κ . It could be a direct economic toll of dollars per day lost from someone not being at work or an indirect economic toll through psychological impacts such as the cost of social services to support isolated people or the overall loss to the economy when they

return to work in a potentially less productive mindset. We have clarified this on page 7.

As our model has many compartments already, we do not further delineate on demographics such as age, mobility, economic status, etc. and therefore it is hard to define a single value of κ . Instead, recognizing κ as the dollar amount per person per day of being in isolation class 2, we can scale the new dimensional cost c by $c = \kappa NC$ where we recognize that κN is the dollar amount per day of every single person in the population being in full isolation. With this definition we see that C , the cost from the original submission, is measured in days. We are now left with cost measured by (2.5) on page 7 which is how the formulation was originally presented.

We highlight now on the bottom of page 7 that once the cost in days is determined, a policy maker need only determine an appropriate value of κ for their region or population to transform back into dollars. Using different examples for κ as mentioned above can give a distribution of possible costs to consider. The reviewer is right to mention that this makes the cost an effective opportunity cost. Indeed, they are correct that because of the lack of detail in partitioning, we are both over and underrepresenting certain demographics. However, we have laid the framework for easy extensions to cost if such a partition was desired; one would need only to define new κ terms and add the group to the cost equation. We thank the reviewer for this thorough analysis and discussion.

7. In regards to the definition of H (P8), you say t_0 starts from when MA begins to strain the healthcare system and then suggest $MA > N_{crit}$ as the system being exceeded and $MA > N_{crit}/2$ is the start of the system being burdened. So it wasn't immediately clear to me if t_0 is when $MA=N_{crit}$ or $N_{crit}/2$.

It should indeed be $N_{crit}/2$, we have clarified this on page 8.

8. Looking at Figure 6 from the appendix, I can't see what's happening in your model. Could you consider removing the baseline and rescaling y_{max} on the left axis so we can see all outbreaks from the blue line?

We have removed the baseline case from all of the graphs of active cases related to η as they are significantly less in most cases than the baseline. We have added legends to all of the requested figures. We have also changed the colourmap on the heatmaps to hopefully be more easily read.

Reviewer 2:

1. This is a good theoretical modelling study of the covid-19 epidemic. The novelty arises from the consideration of the covid-fatigue aspect of social distancing, measured in the model as a cost, considered independently of the direct health costs. I think that this

idea, of a kind of budget for lockdown/distancing patience among the public, is very interesting, and could be explored further.

Thank you, we hope to explore and adapt in future work.

2. One of the problems with this approach is that it is hard to compare the two kinds of costs on the same scale. The authors address this problem by generating plots under different scalings of the costs (ω) (e.g. Fig 7cd). Although this is not entirely satisfactory, it is a step in the right direction. Is it possible to include a graphical representation varying over more values of ω ?

The reviewer is absolutely correct. In this manuscript and any analysis, it is hard to compare economic costs with health and loss-of-life costs. In some sense the parameter ω is determined by how much a society can and is willing to endure high government spending to protect citizens. Our attempt was to just showcase the outcomes at different levels so readers and policy makers can see the impact. The reviewer points out that showing two arbitrary values of ω is probably not that insightful and we agree. Furthermore, realizing that the health cost and social cost figures were just particular values of the total cost when $\omega = 1$ and $\omega = 0$ respectively, we have instead changed our presentation so that we present total cost on a spectrum of ω values from 0 to 1. This is done in Figure 3 on page 11 and Figure 8 on page 16. We hope that this now allows the reader to see how the impacts of behaviours change as weights change and can improve policy decisions. To help with readability of the figures, we have placed a legend in Figures 3 and 8 showing how to interpret increased vigilance and increased spending as values of M_c and C_c .

While not a direct comparison on ω , we have added a new Figure (Figure 5, page 13) which just looks at the health cost ($\omega = 1$) and identifies the appropriate level of vigilance for a given spending (C_c) that will minimize cost. On the same figure we show that in the early days of the pandemic in Ontario, the social distancing was too vigilant which may be contributing to some of the effects of the second wave being experienced now. We think this is a very useful figure.

3. I did not understand the choice in equation 2.6 around the healthcare costs. Specifically why the integral is not taken over the whole duration of the epidemic.

The reviewer brings up a good point, there are many ways to define healthcare costs. Integrating over the whole duration would allow there to be an impact from all case loads of COVID-19 which is very valid. If we did this then it is possible that a scenario with few active cases per day lasting for many years could be as bad or worse than one significantly overloading hospital resources for a number of weeks. In defining H as we have, we are assuming that hospitals and governments would rather sustained outbreaks of few cases than those that exceed the threshold where resources become sparse. While any caseload will have fatalities and other health impacts, we assume that below $N_{\text{crit}}/2$ these are solely due to the nature of the disease and not because of limits in healthcare. Above the threshold however, resources become strained and additional unnecessary

deaths and complications may occur. We have added a discussion about this following (2.7) on page 8. If the cost as we have defined it does not apply to certain regions, the limits of integration can easily be changed, and the costs reanalyzed. To help clarify the choices we have made, we have renamed the healthcare cost throughout the manuscript to be the overburden healthcare cost.

4. My main criticism is that the paper is quite long and the quality of writing is a little inconsistent. This detracts from the clarity of the new ideas that are being described. I recommend that the authors read over the manuscript carefully and try to tighten up the description and presentation of the results.

Perhaps paradoxically, we have increased the length of the paper, but hope that in doing so it has improved the clarity. A lot of the additional length comes from clarifying comments, some new sections in the appendix, and more figures. We think that each of these parts improve the presentation of the manuscript as the reviewer requested. We have removed some of the discussion that did not provide additional meaningful results, analysis, or implications. We have shortened and rewritten most of the results section (bottom of page 9 to bottom of page 14) to better clarify how to read and interpret the heatmaps. For example, rather than list interpretations of several results, we focus mostly on the non-intuitive ones and include graphics of active case studies for each case to demonstrate what is happening. Including more plots of varying ω as the reviewer suggests provides further clarification.

Appendix C

Response to review comments for

Cost and social distancing dynamics in a mathematical model of COVID-19 with application to Ontario, Canada

Authors: I. R. Moyles, J. M. Heffernan, J. D. Kong

submitted to Royal Society Open Science, RSOS-201770R1

We are including a revised version of their article which addresses the minor concerns of Reviewer 1. We have also updated the global COVID-19 case information in the introduction to be relevant as of February 2021.

Reviewer 1:

1. Fig 1 appears to have a red and green line on top of one another. Consider separating these.

The arrows overlap because the compartments can move between each other. We only coloured the arrowheads. We have made this clearer in the text.

2. Every graph with "t (days since March 10)" on the x-axis should be modified to have the date on the x-axis.

Thank you, this has been fixed.

3. Pg 14 point iv. In Australia, an individual who has been identified through contact tracing as somebody who is at risk of having contracted the virus is forced to complete a test. This is virtually the only way an asymptomatic individual case can possibly be identified. Although I can't comment on the capabilities and policies of the Ontario contact tracers, I find it hard to believe that 'curiosity' is a driving factor behind ascertainment of asymptomatic cases.

Certainly in Ontario (and Canada overall) there are anecdotes of people seeking COVID tests for personal ease of mind or otherwise. However, the issue seems to be around the word "curiosity" which we have removed from the manuscript as we feel that "contact tracing or otherwise" adequately captures all cases.

4. Section B (i). The author seems to be suggesting that a series of "plateaus" in the observed data are reflective of an apparent dynamical feed-back loop between incidence and social distancing. A quick look at the daily incidence counts for Ontario reveals a distinct 7-day periodicity from November 1 onwards. Investigation of the "Episode Date" reveals that the best estimate of disease onset is a function of the specimen

collection date. Note also that this periodicity is not present in hospitalisations or deaths.

I suspect that the "plateaus" are actually a feature of how the data was collected.

<https://www.publichealthontario.ca/en/data-and-analysis/infectious-disease/covid-19-data-surveillance/covid-19-data-tool?tab=trends>

Aside from this, I'm not sure what the value of this section is.

The reviewer acknowledges a good point about variability in testing accounting for changes in dynamics. Certainly this is likely a factor, but on its own would not account for the relative week-to-week changes in slope. Perhaps the main issue is that, as the text was worded, it appeared as if we were attempting to capture the feature of the plateaus specifically. This was not the case, instead we were merely trying to demonstrate that a deviation from the data does not invalidate the model itself, but likely the assumptions about the parameters. We have modified the discussion to clarify this and have included the point about the reporting variability (related to the testing rate in our model).

5. Figure 13. I was mainly interested in the social distancing dynamics of your model considering that it is such a significant part of the model. In particular, I believe an important step in your model validation should be to assess how realistic the social distancing dynamics are and I see comparison to mobility data as a useful way of doing this.

I note that social distancing should begin sooner and be more gradual. However, your incidence is well-calibrated within the fitting window.

Indeed, a broader focus on social mobility dynamics and how reliable data could be obtained is a worthwhile endeavour.